# Retrieval-Augmented Diffusion Models for Time Series Forecasting

**Jingwei Liu**[1,2*]   **Ling Yang**[3 †]   **Hongyan Li**[1,2 ‡]   **Shenda Hong**[3,4,5 ‡]

[1]School of Intelligence Science and Technology, Peking University
[2] National Key Laboratory of General Artificial Intelligence, Peking University
[3]Institute of Medical Technology, Peking University Health Science Center
[4] National Institute of Health Data Science, Peking University
[5] Institute for Artificial Intelligence, Peking University
`jingweiliu1996@163.com`, `yangling0818@163.com`
`{leehy, hongshenda}@pku.edu.cn`

## Abstract

While time series diffusion models have received considerable focus from many recent works, the performance of existing models remains highly unstable. Factors limiting time series diffusion models include insufficient time series datasets and the absence of guidance. To address these limitations, we propose a Retrieval-Augmented Time series Diffusion model (RATD). The framework of RATD consists of two parts: an embedding-based retrieval process and a reference-guided diffusion model. In the first part, RATD retrieves the time series that are most relevant to historical time series from the database as references. The references are utilized to guide the denoising process in the second part. Our approach allows leveraging meaningful samples within the database to aid in sampling, thus maximizing the utilization of datasets. Meanwhile, this reference-guided mechanism also compensates for the deficiencies of existing time series diffusion models in terms of guidance. Experiments and visualizations on multiple datasets demonstrate the effectiveness of our approach, particularly in complicated prediction tasks. Our code is available at https://github.com/stanliu96/RATD

## 1   Introduction

Time series forecasting plays a critical role in a variety of applications including weather forecasting [15, 11], finance forecasting [7, 5], earthquake prediction [19] and energy planning [6]. One way to approach time series forecasting tasks is to view them as conditional generation tasks [32, 42], where conditional generative models are used to learn the conditional distribution $P(\boldsymbol{x}^P|\boldsymbol{x}^H)$ of predicting the target time series $\boldsymbol{x}^P$ given the observed historical sequence $\boldsymbol{x}^H$. As the current state-of-the-art conditional generative model, diffusion models [12] have been utilized in many works for time series forecasting tasks [28, 36, 30].

Although the performance of the existing time series diffusion models is reasonably well on some time series forecasting tasks, it remains unstable in certain scenarios (an example is provided in 1(c)). The factors limiting the performance of time series diffusion models are complex, two of them are particularly evident. First, most time series lack direct semantic or label correspondences, which often results in time series diffusion models lacking meaningful **guidance** during the generation

---

*Contact: Jingwei Liu, jingweiliu1996@163.com
†Contributed equally.
‡Corresponding Authors: Hongyan Li, Shenda Hong

process(such as text guidance or label guidance in image diffusion models). This also limits the potential of time series diffusion models.

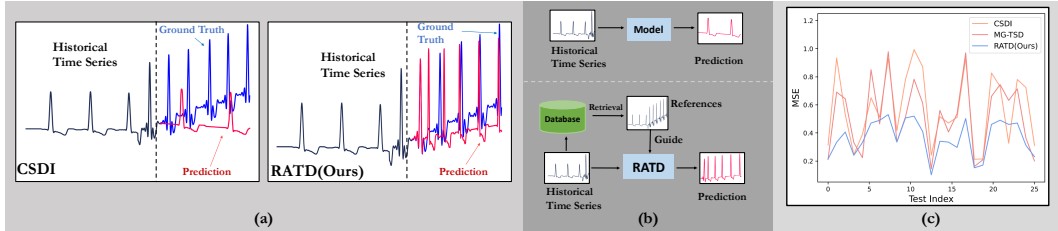

Figure 1: (a) The figure shows the differences in forecasting results between the CSDI [36] (left) and RATD (right). Due to the very small proportion of such cases in the training set, CSDI struggles to make accurate predictions, often predicting more common results. Our method, by retrieving meaningful references as guidance, makes much more accurate predictions. (b) A comparison between our method's framework(bottom) and the conventional time series diffusion model framework(top). (c) We randomly selected 25 forecasting tasks from the electricity dataset. Compared to our method, CSDI and MG-TSD [9] exhibited significantly higher instability. This indicates that the RATD is better at handling complex tasks that are challenging for the other two methods.

The second limiting factor arises from two shortcomings of the time series datasets: **size insufficient** and **imbalanced**. Compared to image datasets, time series datasets typically have a smaller scale. Popular image datasets (such as LAION-400M) contain 400 million sample pairs, while most time series datasets usually only contain tens of thousands of data points. Training a diffusion model to learn the precise distribution of datasets with insufficient size is challenging. Additionally, real-world time series datasets exhibit significant imbalance. For example, in the existing electrocardiogram dataset MIMIC-IV, records related to diagnosed pre-excitation syndrome (PS) account for less than 0.025% of the total records. This imbalance phenomenon may cause models to overlook some extremely rare complex samples, leading to a tendency to generate more common predictions during training, thus making it difficult to handle complex prediction tasks, as illustrated in Figure 1.

To address these limitations, we propose the Retrieval-Augmented Time series Diffusion Model (RATD) for complex time series forecasting tasks. Our approach consists of two parts: the embedding-based retrieval and the reference-guided diffusion model. After obtaining a historical time series, it is input into the embedding-based retrieval process to retrieve the k nearest samples as references. The references are utilized as guidance in the denoising process. RATD focuses on making maximum utilization of existing time series datasets by finding the most relevant references in the dataset to the historical time series, thereby providing meaningful guidance for the denoising process. RATD focuses on maximizing the utilization of insufficient time series data and to some extent mitigates the issues caused by data imbalance. Meanwhile, this reference-guided mechanism also compensates for the deficiencies of guidance in existing time series diffusion models. Our approach demonstrates strong performance across multiple datasets, particularly on more complex tasks.

To summarize, our main contributions are summarized as follows:

- To handle complex time series forecasting, we for the first time introduce Retrieval-Augmented Time series Diffusion (RATD), allowing for greater utilization of the dataset and providing meaningful guidance in the denoising process.

- Extra Reference Modulated Attention (RMA) module is designed to provide reasonable guidance from the reference during the denoising process. RMA effectively simply integrates information without introducing excessive additional computational costs.

- We conducted experiments on five real-world datasets and provided a comprehensive presentation and analysis of the results using multiple metrics. The experimental results demonstrate that our approach achieves comparable or better results compared to baselines.

## 2 Related Work

### 2.1 Diffusion Models for Time Series Forecasting

Recent advancements have been made in the utilization of diffusion models for time series forecasting. In TimeGrad [28], the conditional diffusion model was first employed as an autoregressive approach for prediction, with the denoising process guided by the hidden state. CSDI [36] adopted a non-autoregressive generation strategy to achieve faster predictions. SSSD [1] replaced the noise-matching network with a structured state space model for prediction. TimeDiff [30] incorporated future mix-up and autoregressive initialization into a non-autoregressive framework for forecasting. MG-TSD [9] utilized a multi-scale generation strategy to sequentially predict the main components and details of the time series. Meanwhile, mr-diff [31] utilized diffusion models to separately predict the trend and seasonal components of time series. These methods have shown promising results in some prediction tasks, but they often perform poorly in challenging prediction tasks. We propose a retrieval-augmented framework to address this issue.

### 2.2 Retrival-Augmented Generation

The retrieval-augmented mechanism is one of the classic mechanisms for generative models. Numerous works have demonstrated the benefits of incorporating explicit retrieval steps into neural networks. Classic works in the field of natural language processing leverage retrieval augmentation mechanisms to enhance the quality of language generation [16, 10, 4]. In the domain of image generation, some retrieval-augmented models focus on utilizing samples from the database to generate more realistic images [2, 44]. Similarly, [3] employed memorized similarity information from training data for retrieval during inference to enhance results. MQ-ReTCNN [40] is specifically designed for complex time series forecasting tasks involving multiple entities and variables. ReTime [13] creates a relation graph based on the temporal closeness between sequences and employs relational retrieval instead of content-based retrieval. Although the aforementioned three methods successfully utilize retrieval mechanisms to enhance time series forecasting results, our approach still holds significant advantages. This advantage stems from the iterative structure of the diffusion model, where references can repeatedly influence the generation process, allowing references to exert a stronger influence on the entire conditional generation process.

## 3 Preliminary

The forecasting task and the background knowledge about the conditional time series diffusion model will be discussed in this section. To avoid conflicts, we use the symbol "s" to represent the time series, and the "t" denotes the t-th step in the diffusion process.

**Generative Time Series Forecasting.** Suppose we have an observed historical time series $\boldsymbol{x}^H = \{s_1, s_2, \cdots, s_l \,|\, s_i \in \mathbb{R}^d\}$, where $l$ is the historical time length, $d$ is the number of features per observation and $s_i$ is the observation at time step $i$. The $\boldsymbol{x}^P$ is the corresponding prediction target $\{s_{l+1}, s_{l+2}, \cdots, s_{l+h} \,|\, s_{l+i} \in \mathbb{R}^{d'}\}$ $(d' \leq d)$, where $h$ is the prediction horizon. The task of generative time series forecasting is to learn a density $p_\theta(\boldsymbol{x}^P|\boldsymbol{x}^H)$ that best approximates $p(\boldsymbol{x}^P|\boldsymbol{x}^H)$, which can be written as:

$$\min_{p_\theta} D\left(p_\theta(\boldsymbol{x}^P|\boldsymbol{x}^H)||p(\boldsymbol{x}^P|\boldsymbol{x}^H)\right), \tag{1}$$

where $\theta$ denotes parameters and $D$ is some appropriate measure of distance between distributions. Given observation $x$ the target time series can be obtained directly by sampling from $p_\theta(\boldsymbol{x}^P|\boldsymbol{x}^H)$. Therefore, we obtain the time series $\{s_1, s_2, \cdots, s_{n+h}\} = [\boldsymbol{x}^H, \boldsymbol{x}^P]$.

**Conditional Time Series Diffusion Models.** With observed time series $\boldsymbol{x}^H$, the diffusion model progressively destructs target time series $\boldsymbol{x}_0^P$ (equals to the $\boldsymbol{x}^P$ mentioned in the previous context) by injecting noise, then learns to reverse this process starting from $\boldsymbol{x}_T^P$ for sample generation. For the convenience of expression, in this paper, we use $\boldsymbol{x}_t$ to refer to the t-th time series in the diffusion process, with the letter "P" omitted. The forward process can be formulated as a Gaussian process with a Markovian structure:

$$\begin{aligned} q(\boldsymbol{x}_t|\boldsymbol{x}_{t-1}) &:= \mathcal{N}(\boldsymbol{x}_t; \sqrt{1-\beta_t}\boldsymbol{x}_{t-1}, \boldsymbol{x}^H, \beta_t \boldsymbol{I}), \\ q(\boldsymbol{x}_t|\boldsymbol{x}_0) &:= \mathcal{N}(\boldsymbol{x}_t; \sqrt{\overline{\alpha}_t}\boldsymbol{x}_0, \boldsymbol{x}^H, (1-\overline{\alpha}_t)\boldsymbol{I}), \end{aligned} \tag{2}$$

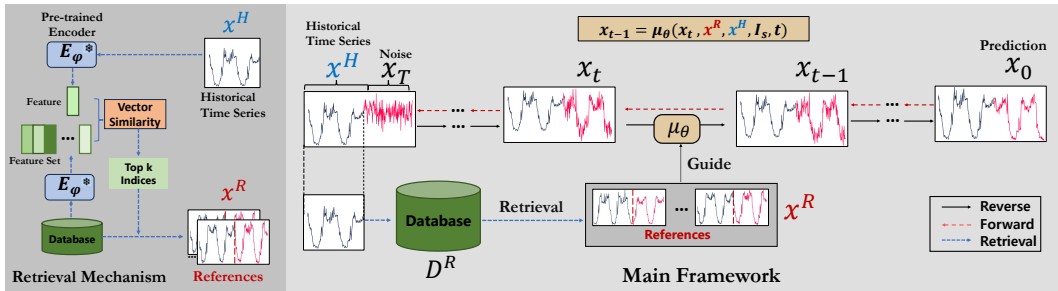

Figure 2: **Overview** of the proposed RATD. The historical time series $\boldsymbol{x}^H$ is inputted into the retrieval module to for the corresponding references $\boldsymbol{x}^R$. After that, $\boldsymbol{x}^H$ is concatenated with the noise as the main input for the model $\mu_\theta$. $\boldsymbol{x}^R$ will be utilized as the guidance for the denoising process.

where $\beta_1, \ldots, \beta_T$ denotes fixed variance schedule with $\alpha_t := 1 - \beta_t$ and $\overline{\alpha}_t := \prod_{s=1}^{t} \alpha_s$. This forward process progressively injects noise into data until all structures are lost, which is well-approximated by $\mathcal{N}(0, \boldsymbol{I})$. The reverse diffusion process learns a model $p_\theta(\boldsymbol{x}_{t-1}|\boldsymbol{x}_t, \boldsymbol{x}^H)$ that approximates the true posterior:

$$p_\theta(\boldsymbol{x}_{t-1}|\boldsymbol{x}_t, \boldsymbol{x}^H) := \mathcal{N}(\boldsymbol{x}_{t-1}; \mu_\theta(\boldsymbol{x}_t), \Sigma_\theta(\boldsymbol{x}_t), \boldsymbol{x}^H), \tag{3}$$

where $\mu_\theta$ and $\Sigma_\theta$ are often computed by the Transformer. Ho *et al.* [12] improve the diffusion training process and optimize following objective:

$$\mathcal{L}(\boldsymbol{x}_0) = \sum_{t=1}^{T} \mathop{\mathbb{E}}_{q(\boldsymbol{x}_t|\boldsymbol{x}_0|\boldsymbol{x}^H)} ||\mu_\theta(\boldsymbol{x}_t, t|\boldsymbol{x}^H) - \hat{\mu}(\boldsymbol{x}_t, \boldsymbol{x}_0|\boldsymbol{x}^H)||^2, \tag{4}$$

where $\hat{\mu}(\boldsymbol{x}_t, \boldsymbol{x}_0|\boldsymbol{x}^H)$ is the mean of the posterior $q(\boldsymbol{x}_{t-1}|\boldsymbol{x}_0, \boldsymbol{x}_t)$ which is a closed from Gaussian, and $\mu_\theta(\boldsymbol{x}_t, t|\boldsymbol{x}^H)$ is the predicted mean of $p_\theta(\boldsymbol{x}_{t-1} \mid \boldsymbol{x}_t|\boldsymbol{x}^H)$ computed by a neural network.

## 4 Method

We first describe the overall architecture of the proposed method in 4.1. Then we will introduce the strategy of building datasets in Section 4.2. The embedding-based retrieval mechanisms and reference-guided time series diffusion model are introduced in Section 4.3.

### 4.1 Framework Overview

Figure 2(a) shows the overall architecture of RATD. We built the entire process based on DiffWave [17], which combines the traditional diffusion model framework and a 2D transformer structure. In the forecasting task, RATD first retrieves motion sequences from the database base $\mathcal{D}^R$ based on the input sequence of historical events. These retrieved samples are then fed into the Reference-Modulated Attention (RMA) as references. In the RMA layer, we integrate the features of the input $[\boldsymbol{x}^H, \boldsymbol{x}^t]$ at time step t with side information $\mathcal{I}_s$ and the references $\boldsymbol{x}^R$. Through this integration, the references guide the generation process. We will introduce these processes in the following subsections.

### 4.2 Constructing Retrieval Database for Time Series

Before retrieval, it is necessary to construct a proper database. We propose a strategy for constructing databases from time series datasets with different characteristics. Some time series datasets are size-insufficient and are difficult to annotate with a single category label (*e.g.*, electricity time series), while some datasets contain complete category labels but exhibit a significant degree of class imbalance (*e.g.*, medical time series). We use two different definitions of databases for these two different types of datasets. For the first definition, the entire training set is directly defined as the database $\mathcal{D}^\mathcal{R}$:

$$\mathcal{D}^\mathcal{R} := \{\boldsymbol{x}_i|\forall \boldsymbol{x}_i \in \mathcal{D}^{\text{train}}\} \tag{5}$$

where $\boldsymbol{x}_i = \{s_i, \cdots, s_{i+l+h}\}$ is the time series with length $l + h$, and $\mathcal{D}^{\text{train}}$ is the training set. In the second way, the subset containing samples from all categories in the dataset is defined as the database

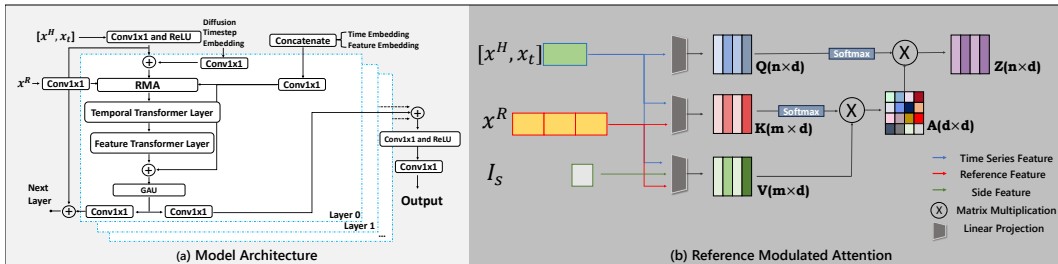

Figure 3: The structure of $\mu_\theta$. (a) The main architecture of $\mu_\theta$ is the time series transformer structure that proved effective. (b) The structure of the proposed RMA. We integrate three different features through matrix multiplication.

$\mathcal{D}^{R'}$:

$$\mathcal{D}^{R'} = \{\boldsymbol{x}_i^c, \cdots, \boldsymbol{x}_q^c | \forall c \in \mathcal{C}\} \tag{6}$$

where $x_i^k$ is the $i$-th sample in the $k$-th class of the training set, with a length of $l + h$. $\mathcal{C}$ is the category set of the original dataset. For brevity, we represent both databases as $\mathcal{D}^R$.

### 4.3 Retrieval-Augmented Time Series Diffusion

**Embedding-Based Retrieval Mechanism** For time forecasting tasks, the ideal references $\{s_i, \cdots, s_{i+h}\}$ would be samples where preceding $n$ points $\{s_{i-n}, \cdots, s_{i-1}\}$ is most relevant to the historical time series $\{s_j, \cdots, s_{j+n}\}$ in the $\mathcal{D}^R$. In our approach, the overall similarity between time series is of greater concern. We quantify the reference between time series using the distance between their embeddings. To ensure that embeddings can effectively represent the entire time series, pre-trained encoders $E_\phi$ are utilized. $E_\phi$ is trained on representation learning tasks, and the parameter set $\phi$ is frozen in our retrieval mechanism. For time series (with length $n + h$) in $\mathcal{D}^R$, their first $n$ points are encoded, thus the $\mathcal{D}^R$ can be represented as $\mathcal{D}_{\text{emb}}^R$:

$$\mathcal{D}_{\text{emb}}^R = \{\{i, E_\phi(\boldsymbol{x}_{[0:n]}^i), \boldsymbol{x}_{[n:n+h]}^i\} | \forall \boldsymbol{x}^i \in \mathcal{D}^R\} \tag{7}$$

where $[p : q]$ refers to the subsequence formed by the $p$-th point to the $q$-th point in the time series. The embedding corresponding to the historical time series can be represented as $\boldsymbol{v}^H = E_\phi(\boldsymbol{x}^H)$. We calculate the distances between $\boldsymbol{v}^H$ and all embeddings in $\mathcal{D}_{\text{emb}}^R$ and retrieve the references corresponding to the $k$ smallest distances. This process can be expressed as:

$$\text{index}(\boldsymbol{v}^H) = \underset{\boldsymbol{x}^i \in \mathcal{D}_{\text{emb}}^R}{\arg\min}^k ||\boldsymbol{v}^H - E_\phi(\boldsymbol{x}_{[0:n]}^i)||^2$$
$$\boldsymbol{x}^R = \{\boldsymbol{x}_{[n:n+h]}^j | \forall j \in \text{index}(\boldsymbol{v}^H)\} \tag{8}$$

where $\text{index}(\cdot)$ represents retrieved index given $\boldsymbol{v}_\mathcal{D}$. Thus, we obtain a subset $\boldsymbol{x}^R$ of $\mathcal{D}^R$ based on a query $\boldsymbol{x}^H$, i.e. $\zeta_k : \boldsymbol{x}^H, \mathcal{D}^R \to \boldsymbol{x}^R$, where $|\boldsymbol{x}^R| = k$.

**Reference-Guided Time Series Diffusion Model** In this section, we will introduce our reference-guided time series diffusion model. In the diffusion process, the forward process is identical to the traditional diffusion process, as shown in Equation (2). Following [34, 12, 35] the objective of the reverse process is to infer the posterior distribution $p(\boldsymbol{z}^{tar}|\boldsymbol{z}^c)$ through the subsequent expression:

$$p(\boldsymbol{x}|\boldsymbol{x}^H) = \int p(\boldsymbol{x}_T|\boldsymbol{x}^H) \prod_{t=1}^T p_\theta(\boldsymbol{x}_{t-1}|\boldsymbol{x}_t, \boldsymbol{x}^H, \boldsymbol{x}^R) \mathcal{D}\boldsymbol{x}_{1:T}, \tag{9}$$

where $p(\boldsymbol{x}_T|\boldsymbol{x}^H) \approx \mathcal{N}(\boldsymbol{x}_T|\boldsymbol{x}^H, \boldsymbol{I})$, $p_\theta(\boldsymbol{x}_{t-1}|\boldsymbol{x}_t, \boldsymbol{x}^H, \boldsymbol{x}^R)$ is the reverse transition kernel from $\boldsymbol{x}_t$ to $\boldsymbol{x}_{t-1}$ with a learnable parameter $\theta$. Following most of the literature in the diffusion model, we adopt the assumption:

$$p_\theta(\boldsymbol{x}_{t-1}|\boldsymbol{x}_t, \boldsymbol{x}) = \mathcal{N}(\boldsymbol{x}_{t-1}; \mu_\theta(\boldsymbol{x}_t, \boldsymbol{x}^H, \boldsymbol{x}^R, t), \Sigma_\theta(\boldsymbol{x}_t, \boldsymbol{x}^H, \boldsymbol{x}^R, t)) \tag{10}$$

where $\mu_\theta$ is a deep neural network with parameter $\theta$. After similar computations as those in [12], $\Sigma_\theta(\boldsymbol{x}_t, \boldsymbol{x}^H, \boldsymbol{x}^R, t)$ in the backward process is approximated as fixed. In other words, we can achieve reference-guided denoising by designing a rational and robust $\mu_\theta$.

**Denoising Network Architecture**   Similar to DiffWave [17] and CSDI [36], our pipeline is constructed on the foundation of transformer layers, as shown in Figure 3. However, the existing framework cannot effectively utilize the reference as guidance. Considering attention modules to integrate the $\boldsymbol{x}^R$ and $\boldsymbol{x}_t$ as a reasonable intuition, we propose a novel module called Reference Modulated Attention (RMA). Unlike normal attention modules, we realize the fusion of three features in RMA: the current time series feature, the side feature, and the reference feature. To be specific, RMA was set at the beginning of each residual module Figure 3. We use 1D-CNN to extract features from the input $\boldsymbol{x}_t$, references $\boldsymbol{x}^R$, and side information. Notably, we concatenate all references together for feature extraction. Side information consists of two parts, representing the correlation between variables and time steps in the current time series dataset Appendix B. We adjust the dimensions of these three features with linear layers and fuse them through matrix dot products. Similar to text-image diffusion models [29], RMA can effectively utilize reference information to guide the denoising process, while appropriate parameter settings prevent the results from overly depending on the reference.

**Training Procedure**   To train RATD (*i.e.*, optimize the evidence lower bound induced by RATD), we use the same objective function as previous work. The loss at time step $t - 1$ are defined as follows respectively:

$$
\begin{aligned}
L_{t-1}^{(x)} &= \frac{1}{2\tilde{\beta}_t^2}\|\mu_\theta(\boldsymbol{x}_t, \hat{\boldsymbol{x}}_0) - \hat{\mu}(\boldsymbol{x}_t, \hat{\boldsymbol{x}}_0)\|^2 \\
&= \gamma_t\|\boldsymbol{x}_0 - \hat{\boldsymbol{x}}_0\|
\end{aligned}
\tag{11}
$$

where $\hat{\boldsymbol{x}}_0$ are predicted from $\boldsymbol{x}_t$ , and $\gamma_t = \frac{\bar{\alpha}_{t-1}\beta_t^2}{2\tilde{\beta}_t^2(1-\bar{\alpha}_t)^2}$ are hyperparameters in diffusion process. We summarize the training procedure of RATD in Algorithm 1 and highlight the differences from the conventional models, in cyan. The process of sampling is shown in Appendix A.

---

**Algorithm 1** Training Procedure of RATD

---

**Require:** Time series dataset $\mathcal{D}^{\text{train}}$, neural network $\mu_\theta$, , diffusion step $T$, external database $\mathcal{D}^R$, pre-trained encoder $E_\phi$, number of references $k$
  1: Retrieve references with top-$k$ high similarity from $\mathcal{D}^R$ using $E$ to obtain $\boldsymbol{x}^R$ as described in Section 4.3
  2: **while** $\phi_\theta$ not converge **do**
  3:     Sample diffusion time $t \in \mathcal{U}(0, \ldots, T)$
  4:     Compute the side feature $\mathcal{I}_s$
  5:     Perturb $\boldsymbol{x}_0$ to obtain $\boldsymbol{x}_t$
  6:     Predict $\hat{\boldsymbol{x}}_0$ from $\boldsymbol{x}_t$, $\mathcal{I}_s$ and $\boldsymbol{x}^R$                          (Equation (10))
  7:     Compute loss $L$ with $\hat{\boldsymbol{x}}_0$ and $\boldsymbol{x}_0$                                    (Equation (11))
  8:     Update $\theta$ by minimizing $L$
  9: **end while**

---

## 5   Experiments

### 5.1   Experimental Setup

**Datasets**  Following previous work [45, 38, 8, 30], experiments are performed on four popular real-world time series datasets: (1) *Electricity*[*], which includes the hourly electricity consumption data from 321 clients over two years.; (2) *Wind* [20], which contains wind power records from 2020-2021. (3) *Exchange* [18], which describes the daily exchange rates of eight countries (Australia, British, Canada, Switzerland, China, Japan, New Zealand, and Singapore); (4) *Weather*[†], which documents 21 meteorological indicators at 10-minute intervals spanning from 2020 to 2021.; Besides, we also applied our method to a large ECG time series dataset: MIMIC-IV-ECG [14]. The MIMIC-IV-ECG dataset contains clinical electrocardiogram data from over 190,000 patients and 450,000 hospitalizations at Beth Israel Deaconess Medical Center (BIDMC).

---

[*]https://archive.ics.uci.edu/ml/datasets/ElectricityLoadDiagrams20112014
[†]https://www.bgc-jena.mpg.de/wetter/

Table 1: Performance comparisons on four real-world datasets in terms of MSE, MAE, and CRPS. The best is in bold, while the second best is underlined.

| Dataset | Exchange | | | Wind | | | Electricity | | | Weather | | |
|---|---|---|---|---|---|---|---|---|---|---|---|---|
| Metric | MSE | MAE | CRPS | MSE | MAE | CRPS | MSE | MAE | CRPS | MSE | MAE | CRPS |
| **RATD (ours)** | **0.013** | **0.073** | **0.339** | **0.784** | **0.579** | **0.673** | 0.151 | 0.246 | **0.373** | **0.281** | **0.293** | **0.301** |
| TimeDiff | 0.018 | 0.091 | 0.589 | 0.896 | 0.687 | 0.917 | 0.193 | 0.305 | 0.490 | 0.327 | 0.312 | 0.410 |
| CSDI | 0.077 | 0.194 | 0.397 | 1.066 | 0.741 | 0.941 | 0.379 | 0.579 | 0.480 | 0.356 | 0.374 | 0.354 |
| mr-Diff | 0.016 | 0.082 | 0.397 | 0.881 | 0.675 | 0.881 | 0.173 | 0.258 | 0.429 | 0.296 | 0.324 | 0.347 |
| D$_3$VAE | 0.200 | 0.301 | 0.401 | 1.118 | 0.779 | 0.979 | 0.286 | 0.372 | 0.389 | 0.315 | 0.380 | 0.381 |
| Fedformer | 0.133 | 0.233 | 0.631 | 1.113 | 0.762 | 1.235 | 0.238 | 0.341 | 0.561 | 0.342 | 0.347 | 0.319 |
| FreTS | 0.039 | 0.140 | 0.440 | 1.004 | 0.703 | 0.943 | 0.269 | 0.371 | 0.634 | 0.351 | 0.354 | 0.391 |
| FiLM | 0.016 | 0.079 | 0.349 | 0.984 | 0.717 | 0.798 | 0.210 | 0.320 | 0.671 | 0.327 | 0.336 | 0.556 |
| iTransformer | 0.016 | 0.074 | 0.343 | 0.932 | 0.676 | 0.811 | 0.192 | 0.262 | 0.402 | 0.358 | 0.401 | 0.318 |
| Autoformer | 0.056 | 0.167 | 0.769 | 1.083 | 0.756 | 1.201 | 1.026 | 0.313 | 0.602 | 0.360 | 0.354 | 0.754 |
| Pyraformer | 0.032 | 0.112 | 0.532 | 1.061 | 0.735 | 0.994 | 0.273 | 0.379 | 0.732 | 0.394 | 0.385 | 0.485 |
| Informer | 0.073 | 0.192 | 0.631 | 1.168 | 0.772 | 1.065 | 0.292 | 0.383 | 0.749 | 0.385 | 0.364 | 0.821 |
| PatchTST | 0.047 | 0.153 | 0.629 | 1.001 | 0.672 | 1.026 | 0.225 | 0.394 | 0.801 | 0.782 | 0.670 | 0.370 |
| SCINet | 0.038 | 0.137 | 0.624 | 1.055 | 0.732 | 0.997 | 0.171 | 0.280 | 0.499 | 0.329 | 0.344 | 0.814 |
| DLinear | 0.022 | 0.102 | 0.538 | 0.899 | 0.686 | 0.957 | 0.215 | 0.336 | 0.527 | 0.488 | 0.444 | 0.791 |
| NLinear | 0.019 | 0.091 | 0.481 | 0.989 | 0.706 | 0.974 | 0.147 | **0.239** | 0.419 | 0.369 | 0.328 | 0.738 |
| TimesNet | 0.023 | 0.120 | 0.520 | 0.982 | 0.771 | 1.001 | **0.141** | 0.361 | 0.403 | 0.313 | 0.364 | 0.491 |
| NBeats | 0.016 | 0.081 | 0.399 | 1.069 | 0.741 | 0.981 | 0.269 | 0.370 | 0.697 | 0.744 | 0.420 | 0.871 |

**Baseline Methods** To comprehensively demonstrate the effectiveness of our method, we compare RATD with four kinds of time series forecasting methods. Our baselines include (1) Time series diffusion models, including CSDI [36], mr-Diff [31], D$^3$VAE [20], TimeDiff [30]; (2) Recent time series forecasting methods with frequency information, including FiLM [46], Fedformer [47] and FreTS [41] ; (3) Time series transformers, including PatchTST [25], Autoformer [38], Pyraformer [22], Informer [45] and iTransformer [23]; (4) Other popular methods, including TimesNet [39], SciNet [21], Nlinear [43], DLinear [43] and NBeats [26].

**Evaluation Metric** To comprehensively assess our proposed methodology, our experiment employs three metrics: (1) Probabilistic forecasting metrics: Continuous Ranked Probability Score (CRPS) on each time series dimension [24]. (2) Distance metrics: Mean Squared Error (MSE), and Mean Average Error(MAE) are employed to measure the distance between predictions and ground truths.

**Implementation Details** The length of the historical time series was 168, and the prediction lengths were (96, 192, 336), with results averaged. All experiments were conducted on an Nvidia RTX A6000 GPU with 40GB memory. During the experiments, the second strategy of conducting $\mathcal{D}^R$ was employed for the MIMIC dataset, while the first strategy was utilized for the other four datasets. To reduce the training cost, we preprocessed the retrieval process by storing the reference indices of each sample in the training set in a dictionary. During the training on the diffusion model, we accessed this dictionary directly to avoid redundant retrieval processes. More details are shown in Appendix B.

## 5.2 Main Results

Table 1 presents the primary results of our experiments on four daily datasets. Our approach surpasses existing time series diffusion models. Compared to other time series forecasting methods, our approach exhibits superior performance on three out of four datasets, with competitive performance on the remaining dataset. Notably, we achieve outstanding results on the wind dataset. Due to the lack of clear short-term periodicity (daily or hourly), some prediction tasks in this dataset are exceedingly challenging for other models. Retrieval-augmented mechanisms can effectively assist in addressing these challenging prediction tasks.

Figure 4 presents a case study randomly selected from our experiments on the wind dataset. We compare our prediction with iTransformer and two popular open-source time series diffusion models, CSDI and D$_3$VAE. Although CSDI and D$_3$VAE provide accurate predictions in the initial short-term period, their long-term predictions deviate significantly from the ground truth due to the lack of guidance. ITransformer captures rough trends and periodic patterns, yet our method offers higher-quality predictions than the others. Furthermore, through the comparison between the predicted

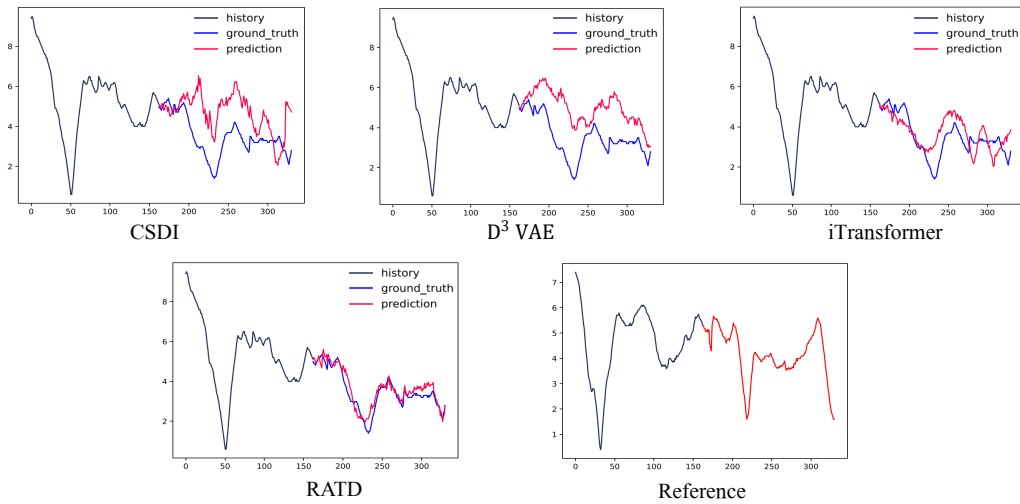

Figure 4: Visualizations on *wind* by CSDI, $D_3$VAE, iTransformer and the proposed RATD (with reference).

results and references in the figure, although references provide strong guidance, they do not explicitly substitute for the entire generated results. This further validates the rationality of our approach.

Table 2 presents the testing results of our method on the MIMIC-IV-ECG dataset. We selected some powerful open-source methods as baselines for comparison. Our experiments are divided into two parts: in the first part, we evaluate the entire test set, while in the second part, we select rare cases (those accounting for less than 2% of total cases) from the test set as a subset for evaluation. Prediction tasks in the second part are more challenging for deep models. In the first experiment, our method achieved results close to iTransformer, while in the second task, our model significantly outperformed other methods, demonstrating the effectiveness of our approach in addressing challenging tasks.

## 5.3 Model Analysis

**Influence of Retrieval Mechanism**    To investigate the impact of the retrieval augmentation mechanism on the generation process, we conducted an ablation study and presented the results in Table 3. The study addresses two questions: whether the retrieval augmentation mechanism is effective and which retrieval method is most effective. Firstly, we removed our retrieval augmentation mechanism from the RATD as a baseline. Besides, the model with random time series guidance is another baseline. The references retrieved by other methods have all positively impacted the prediction results. This suggests that reasonable references are highly effective in guiding the generation process.

We also compared two different retrieval mechanisms: correlation-based retrieval and embedding-based retrieval. The first method directly retrieves the reference in the time domain (*e.g.*, using Dynamic Time Warping (DTW) or Pearson correlation coefficient). Our approach adopts the second mechanism: retrieving references through the embedding of time series. From the results, the correlation-based methods are significantly inferior to the embedding-based methods. The former methods fail to capture the key features of the time series, making it difficult to retrieve the best references for forecasting. We also evaluate the embedding-based methods with various encoders for comparison. The comprehensive results show that methods with different encoders do not significantly differ. This indicates that different methods can all extract meaningful references, thereby producing similar improvements in results. TCN was utilized in our experiment because TCN strikes the best balance between computational cost and performance.

**Effect of Retrieval Database**    We conducted an ablation study on two variables, $n$ and $k$, to investigate the influence of the retrieval database $\mathcal{D}^R$ in RATD, where $n$ represents the number of samples in each category of the database, and $k$ represents the number of reference exemplars. The results in Figure 5q can benefit the model in terms of prediction accuracy because a larger $\mathcal{D}^R$ brings higher diversity, thereby providing more details beneficial for prediction and enhancing the generation

Table 2: Performance comparisons on MIMIC datasets with popular time series forecasting methods. Here, "MIMIC-IV (All)" refers to the model's testing results on the complete test set, while "MIMIC(Rare)" indicates the model's testing results on a rare disease subset.

| Method | iTransformer | | | PatchTST | | | TimesNet | | | CSDI | | | RATD | | |
|---|---|---|---|---|---|---|---|---|---|---|---|---|---|---|---|
| Metric | MSE | MAE | CRPS | MSE | MAE | CRPS | MSE | MAE | CRPS | MSE | MAE | CRPS | MSE | MAE | CRPS |
| MIMIC-IV (All) | 0.174 | **0.263** | 0.299 | 0.219 | 0.301 | 0.307 | 0.193 | 0.311 | 0.310 | 0.268 | 0.331 | 0.369 | **0.172** | 0.270 | **0.293** |
| MIMIC-IV (Rare) | 0.423 | 0.315 | 0.379 | 0.483 | 0.379 | 0.407 | 0.627 | 0.359 | 0.464 | 0.499 | 0.359 | 0.374 | **0.206** | **0.299** | **0.301** |

Table 3: Ablation study on different retrieval mechanisms. "-" means no references was utilized and "Random" means references are selected randomly. Others refer to what model we use for retrieval references.

| Dataset | Exchange | | | Wind | | | Electricity | | | Weather | | |
|---|---|---|---|---|---|---|---|---|---|---|---|---|
| Metric | MSE | MAE | CRPS | MSE | MAE | CRPS | MSE | MAE | CRPS | MSE | MAE | CRPS |
| - | 0.077 | 0.194 | 0.397 | 1.066 | 0.741 | 0.941 | 0.379 | 0.579 | 0.480 | 0.356 | 0.374 | 0.354 |
| Random | 0.153 | 0.203 | 0.599 | 1.593 | 0.903 | 0.996 | 0.471 | 0.639 | 0.701 | 0.431 | 0.473 | 0.461 |
| DTW | 0.075 | 0.195 | 0.403 | 1.073 | 0.791 | 0.942 | 0.357 | 0.564 | 0.449 | 0.361 | 0.375 | 0.356 |
| Pearson | 0.091 | 0.207 | 0.411 | 1.099 | 0.831 | 0.953 | 0.361 | 0.571 | 0.483 | 0.370 | 0.364 | 0.391 |
| DLinear | 0.022 | 0.081 | 0.361 | 0.941 | 0.735 | 0.895 | **0.159** | **0.250** | **0.390** | 0.297 | 0.304 | 0.332 |
| Informer | 0.019 | 0.078 | 0.371 | 0.841 | 0.645 | 0.861 | 0.170 | 0.263 | 0.411 | 0.291 | 0.305 | 0.330 |
| TimesNet | **0.013** | 0.074 | 0.341 | **0.781** | **0.572** | **0.669** | 0.167 | 0.263 | 0.397 | 0.286 | 0.295 | **0.311** |
| TCN | **0.013** | **0.073** | **0.339** | 0.784 | 0.579 | 0.673 | 0.161 | 0.256 | 0.391 | **0.281** | **0.293** | 0.313 |

process. Simply increasing k does not show significant improvement, as utilizing more references may introduce more noise into the denoising process. In our experiment, the settings of $n$ and $k$ are 256 and 3, respectively.

**Inference Efficiency**   In this experiment, we evaluate the inference efficiency of the proposed RATD in comparison to other baseline time series diffusion models (TimeGrad, MG-TSD, SSSD). Figure 6 illustrates the inference time on the multivariate *weather* dataset with varying values of the prediction horizon ($h$). While our method introduces an additional retrieval module, the sampling efficiency of the RATD is not low due to the non-autoregressive transformer framework. It even slightly outperforms other baselines across all $h$ values. Notably, TimeGrad is observed to be the slowest, attributed to its utilization of auto-regressive decoding.

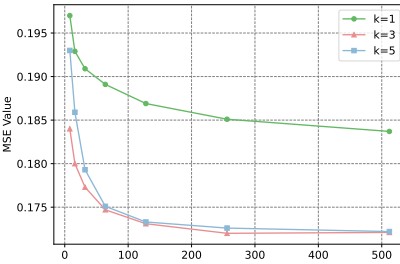
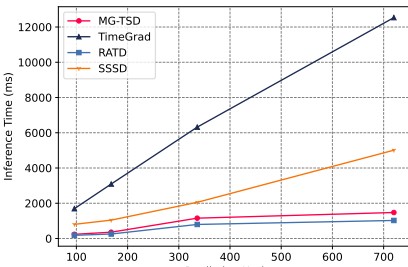

Figure 5: The effect of hyper-parameter $n$ and $k$.

Figure 6: Inference time (ms) on the Electricity with different prediction horizon $h$

**Effectiveness of Reference Modulated Attention**   To validate the effectiveness of the proposed RMA, we designed additional ablation experiments. In these experiments, we used the CSDI architecture as the baseline method and added extra fusion modules to compare the performance of these modules (linear layer, cross-attention layer, and RMA). The results are shown in the Table 4.

Through our experiments, we found that compared to the basic cross-attention-based approach, RMA can integrate an edge information matrix (representing correlations between time and feature dimensions) more effectively. The extra fusion is highly beneficial in experiments, guiding the model to capture relationships between different variables. In contrast, linear-based methods concatenate inputs and references initially, which prevents the direct extraction of meaningful information from references, resulting in comparatively modest performance.

Table 4: Performance comparison(MSE) between CSDI-based methods, CSDI represents the basic network framework, CSDI+Linear denotes the approach where inputs and references are concatenated via a linear layer and fed into the network together, CSDI+CrossAttention signifies the use of cross attention to fuse features from inputs and references, and finally, CSDI+RMA, which incorporates an additional RMA.

| Dataset | Exchange | Electricity | Wind | Weather | Solar | MIMIC-IV |
|---|---|---|---|---|---|---|
| CSDI | 0.077 | 0.379 | 1.066 | 0.356 | 0.381 | 0.268 |
| CSDI+Linear | 0.075 | 0.316 | 0.932 | 0.349 | 0.369 | 0.265 |
| CSDI+Cross Attention | 0.028 | 0.173 | 0.829 | 0.291 | 0.340 | 0.183 |
| CSDI+RMA | **0.013** | **0.151** | **0.784** | **0.281** | **0.327** | **0.172** |

**Predicting $x_0$ vs Predicting $\epsilon$.** Following the formulation in Section 4.3, our network is designed to forecast the latent variable $x_0$. Since some existing models [28, 36] have been trained by predicting an additional noise term $\epsilon$, we conducted a comparative experiment to determine which approach is more suitable for our framework. Specifically, we maintained the network structure unchanged, only modifying the prediction target to be $\epsilon$. The results are presented in Table 5. Predicting $x_0$ proves to be more effective. This may be because the relationship between the reference and $x_o$ is more direct, making the denoising task relatively easier.

Table 5: MSEs of two denoising strategies: Predicting $x_0$ vs predicting $\epsilon$.

| denoising strategy | Wind | Weather | Exchange |
|---|---|---|---|
| $x_0$ | **0.784** | **0.281** | **0.013** |
| $\epsilon$ | 0.841 | 0.331 | 0.018 |

**RMA position** We investigate the best position of RMA in the model. Front, middle, and back means we set the RMA in the front of, in the middle of, and the back of two transformer layers, respectively. We found that placing RMA before the bidirectional transformer resulted in the most significant improvement in model performance. This also aligns with the intuition of network design: cross-attention modules placed at the front of the model tend to have a greater impact.

Table 6: Ablation study on different RMA positions. The best is in bold.

| Dataset | Exchange | | | Wind | | | Electricity | | | Weather | | |
|---|---|---|---|---|---|---|---|---|---|---|---|---|
| Metric | MSE | MAE | CRPS | MSE | MAE | CRPS | MSE | MAE | CRPS | MSE | MAE | CRPS |
| - | 0.077 | 0.194 | 0.397 | 1.066 | 0.741 | 0.941 | 0.379 | 0.579 | 0.480 | 0.356 | 0.374 | 0.354 |
| Back | 0.031 | 0.105 | 0.373 | 0.673 | 0.611 | 0.842 | 0.267 | 0.434 | 0.426 | 0.301 | 0.321 | 0.322 |
| Middle | 0.057 | 0.141 | 0.381 | 0.799 | 0.631 | 0.833 | 0.291 | 0.481 | 0.451 | 0.333 | 0.331 | 0.336 |
| Front | **0.013** | **0.063** | **0.331** | **0.784** | **0.579** | **0.673** | **0.161** | **0.256** | **0.391** | **0.281** | **0.293** | **0.313** |

# 6 Discussion

**Limitation and Future Work** As a transformer-based diffusion model structure, our approach still faces some challenges brought by the transformer framework. Our model consumes a significant amount of computational resources dealing with time series consisting of too many variables. Additionally, our approach requires additional preprocessing (retrieval process) during training, which incurs additional costs on training time (around ten hours).

**Conclusion** In this paper, we propose a new framework for time series diffusion modeling to address the forecasting performance limitations of existing diffusion models. RATD retrieves samples most relevant to the historical time series from the constructed database and utilize them as references to guide the denoising process of the diffusion model, thereby obtaining more accurate predictions. RATD is highly effective in solving challenging time series prediction tasks, as evaluated by experiments on five real-world datasets.

# Acknowledgements

This work is supported by the National Natural Science Foundation of China (No.62172018, No.62102008) and Wuhan East Lake High-Tech Development Zone National Comprehensive Experimental Base for Governance of Intelligent Society.

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

# A    Sampling Procedure

Like Algorithm 1, we summarize the sampling procedure of RATD in Algorithm 2 and highlight the differences from conventional diffusion models in .

---

**Algorithm 2** Sampling Procedure of RATD

---

**Require:** The historical time series $x^H$, the learned model $\mu_\theta$, external database $\mathcal{D}^R$, pre-trained $E_\phi$, the number of references $k$

**Ensure:** Prediction $x^P$ corresponding to the history $x^H$
  1: Sample initial target time series $x_T$
  2: Embed $x^H$ into $v^H$
  3: Retrieval the reference $x^R$ with $v^H$
  4: Compute the side feature $\mathcal{I}_s$
  5: **for** $t$ in $T, T-1, \ldots, 1$ **do**
  6:     Predict $\hat{x}_0$ from $x_t$, $\mathcal{I}_s$ and $x^R$          (Equation (10))
  7:     Sample $x_{t-1}$ from the posterior $q(x_t|x_0)$          (Equation (2))
  8: **end for**

---

# B    Impletion Details

## B.1    Training Details

Our dataset is split in the proportion of 7:1:2 (Train: Validation: Test), utilizing a random splitting strategy to ensure diversity in the training set. We sample the ECG signals at 125Hz for the MIMIC-IV dataset and extract fixed-length windows as samples. For training, we utilized the Adam optimizer with an initial learning rate of $10^{-3}$, $betas = (0.95, 0.999)$. During the training process of shifted diffusion, the batch size was set to 64, and early stopping was applied for a maximum of 200 epochs. The diffusion steps $T$ were set to 100.

## B.2    Side Information

We combine temporal embedding and feature embedding as side information $v_s$. We use 128-dimensions temporal embedding following previous studies [37]:

$$s_{embedding}(s_\zeta) = \left( \sin(s_\zeta/\tau^{0/64}), \ldots, \sin(s_\zeta/\tau^{63/64}), \cos(s_\zeta/\tau^{0/64}), \ldots, \cos(s_\zeta/\tau^{63/64}) \right) \quad (12)$$

where $\tau = 10000$. Following [36], $s_l$ represents the timestamp corresponding to the l-th point in the time series. This setup is designed to capture the irregular sampling in the dataset and convey it to the model. Additionally, we utilize learnable embedding to handle feature dimensions. Specifically, feature embedding is represented as 16-dimensional learnable vectors that capture relationships between dimensions. According to [17], we combine time embedding and feature embedding, collectively referred to as side information $\mathcal{I}_s$.

The shape of $\mathcal{I}_s$ is not fixed and varies with datasets. Taking the Exchange dataset as an example, the shape of forecasting target $x^R$ is [Batchsize (64), 7(number of variables), 168 (time-dimension), 12 (time-dimension)] and the corresponding shape of $\mathcal{I}_s$ is [Batchsize (64), total channel(144( time:128 + feature:16)), 320 (frequency-dimension*latent channel), 12 (time-dimension)].

## B.3    Transformers Details

Our approach employs the Transformer architecture from CSDI, with the distinction of expanding the channel dimension to 128. The network comprises temporal and feature layers, ensuring the comprehensiveness of the model in handling the time-frequency domain latent while maintaining a relatively simple structure. Regarding the transformer layer, we utilized a 1-layer Transformer encoder implemented in PyTorch [27], comprising multi-head attention layers, fully connected layers, and layer normalization. We adopted the "linear attention transformer" package [‡], to enhance

---

[‡]`https://github.com/lucidrains/linear-attention-transformer`

computational efficiency. The inclusion of numerous features and long sequences prompted this decision. The package implements an efficient attention mechanism [33], and we exclusively utilized the global attention feature within the package.

## B.4 Metrics

We will introduce the metrics in our experiments. We summarize them as below:

**CRPS.** CRPS [24] is a univariate strictly proper scoring rule which ´ measures the compatibility of a cumulative distribution function $F$ with an observation $x$ as:

$$CRPS(F, x) = \int_R (F(y) - \mathbb{1}_{(x \leq y)})^2 dy \tag{13}$$

where $\mathbb{1}_{(x \leq y)}$ is the indicator function, which is 1 if $x \leq y$ and 0 otherwise. The CRPS attains the minimum value when the predictive distribution $F$ same as the data distribution.

**MAE and MSE.** MAE and MSE are calculated in the formula below, $\hat{\boldsymbol{x}}^P$ represents the predicted time series, and $\boldsymbol{x}^P$ represents the ground truth time series. MAE calculates the average absolute difference between predictions and true values, while MSE calculates the average squared difference between predictions and true values. A smaller MAE or MSE implies better predictions.

$$MAE = mean(|\hat{\boldsymbol{x}}^P - \boldsymbol{x}^P|)$$
$$MSE = \sqrt{mean(|\hat{\boldsymbol{x}}^P - \boldsymbol{x}^P|)} \tag{14}$$

