# OpenReview forum: "Retrieval-Augmented Diffusion Models for Time Series Forecasting"
_NeurIPS.cc/2024/Conference — NeurIPS 2024 poster_

### Official Review · Reviewer_zh2V · 2024-07-10

**Soundness:** 3
**Presentation:** 3
**Contribution:** 2
**Rating:** 6
**Confidence:** 3

**Summary:**

This paper proposed Retrieval-Augmented Time series Diffusion (RATD) model for complex time series forecasting tasks, and designed an Extra Reference Modulated Attention (RMA) module to enable the guidance from the retrieved reference time series during the denoising process of the diffusion models. On five real-world datasets, this paper demonstrated that RATD performed on-par or better on multiple metrics compared to various time series forecasting baselines including diffusion models, transformers, and other methods.

**Strengths:**

1. This paper is well-written and easy to follow.
2. The methodology design is clear and reasonable to me.
3. The experiments are comprehensive, and demonstrated the effectiveness of RATD across various retrieval mechanisms on multiple time series datasets using multiple metrics.
4. In the experiment results, RATD is effective with k = 3 retrieved reference time series, and the inference time is also on the lower side compared to other baselines, which showed that RATD should be relatively computationally scalable and efficient.

**Weaknesses:**

1. Based on the related work section, there have been several works which investigated  time series forecasting using diffusion models and achieved promising results. RAG is also a well-studied topic in text generation. Given this, I think the technical novelty of this work is moderate, though it's a nice direction to combine RAG with time series forecasting and the experiment results demonstrated its effectiveness. I am wondering whether there is any previous work on time series RAG?
2. Probably out of the scope of this paper, it would be great if the authors can provide some discussion or intuitive insights on the advantage of using diffusion models for time series forecasting compared to other methods, e.g. is it more robust or generalizable, and also the advantage of diffusion models when it comes to retrieval augmented forecasting. As in table 1, several transformer based methods also performed well on some datasets, thus I am wondering whether there is any justification in choosing diffusion models for time series forecasting given its potential extra computational cost.

**Questions:**

1. In section 4.2, how is $D^{R}$ different from $D^{R'}$ if $\mathcal{C}$ represents all categories? In 5.3, it seems that when it's $D^{R'}$, only $n$ samples from each category are used to construct the database. Is $n$ a hyperparameter for $D^{R'}$ that also needs to be introduced in section 4.2?
2. In section 4.3, under $\textbf{Denoising Network Architecture}$, it is mentioned that all references are concatenated together for feature extraction. I am wondering how this scales with the length of time series, i.e. would there be memory concerns if you are working with long-horizon time series forecasting / long context windows?

**Limitations:**

The authors discussed the limitations of this work with regards to computational costs incurred by the transformer based diffusion model and the retrieval process, which is also my concern.

---

> ### Author Rebuttal · Authors · 2024-08-06
>
> *Thank you very much to reviewer zh2V for the careful reading and consideration, as well as for acknowledging the innovative aspects of our approach* .
>
> **Q1**: I am wondering whether there is any previous work on time series RAG?
>
> **A1**: In the related work section, I mentioned that previous studies have combined time series prediction with RAG[1][2]. Our proposed method differs from these prior works in several key aspects:
>
> **Different Utilization of References**: Previous methods either directly fused features using existing cross-attention modules or concatenated them for fusion. In contrast, we introduce a novel RMA(Reference Modulated Attention) module for feature fusion.
>
> **Distinct Research Motivation**: We believe that time series diffusion models may offer advantages over transformer networks in certain prediction tasks. However, these advantages are not always pronounced (as mentioned in your subsequent question). We identify the lack of guidance mechanisms as a major barrier to achieving greater performance with diffusion models. Due to this absence, diffusion models must directly learn the mapping from Gaussian distribution to data distribution, which can be challenging. We propose an enhanced retrieval-guided mechanism and aim to provide new insights for future research on time series diffusion models, particularly in the study of guidance mechanisms.
>
> **Differences in Experimental Completeness**: Previous studies did not provide comprehensive evaluations of public datasets, or experiment details, making it difficult to assess model performance. for the sake of comprehensive experimental comparison, we present here the MSE results of our replicated method under our experimental settings, demonstrating that our approach shows noticeable advantages.
> |Dataset |Exchange| Wind | Traffic | Weather|
> | :-----| :---- | :---- |:---- |:---- |
> |RATD（Ours）  | 0.013(0.001)  | 0.784(0.005)  | 0.151(0.002) |0.281(0.002) |
> |MQRetNN[1]| 0.063(0.004) | 1.116(0.008) |0.346(0.003) |0.668(0.004) |
> |ReTime[2]| 0.059(0.003) | 1.043(0.007) |0.330(0.005)|0.489(0.005)|
>
> Clearly, RATD exhibits significant performance advantages, indicating that our method can better utilize references for prediction.
>
> In summary, while there are similarities between prior works and our method to some extent, there are significant differences as well. We believe our approach advances the integration of RAG with time series prediction tasks.
>
> **Q2**:  I am wondering whether there is any justification in choosing diffusion models for time series forecasting given its potential extra computational cost.
>
> **A2**: The diffusion model is a generative framework and the transformer is a specific model structure. Two methods follow different technical paths, making it challenging to assess their relative performance. To be specific, many existing efforts focus on enhancing performance by updating the architecture of transformers [3], while related works on diffusion models concentrate on updating the framework as a whole [4].  However, I believe time series diffusion models hold greater research potential.
> Firstly, time series diffusion models are currently constrained by inference costs, preventing the direct adoption of powerful time series transformer models as their denoising network. In the future, if transformer models with significantly lower inference costs and high performance emerge, the performance of diffusion models could have substantial improvements.
> Secondly, current time series diffusion models lack clear and effective guidance mechanisms. This work is among the few that address this issue, and I believe that appropriate guidance mechanisms could greatly enhance the performance of diffusion models (as demonstrated in the experimental section of this paper). Further research into guidance mechanisms may activate the potential of time series diffusion models in the future.
>
> **Q3**: In section 4.2, how is DR different from DR′ if C represents all categories? Is n a hyperparameter for DR′ that also needs to be introduced in section 4.2?
>
> **A3**: In fact, we employed two different methods for constructing retrieval datasets, primarily to correspond to two types of time series datasets: one smaller and lacking clear category labels, such as wind, electric...; and one with complete category annotations, often larger, such as MIMIC-IV. For the former, we could only use the first method to construct the retrieval dataset, while for the latter, both methods were applicable. If all category data from the latter case were applied, as you mentioned, then there would be no difference between the two construction methods.
>
> Additionally, you are correct in noting that the sample counts for each category should also be included in the formula. Thank you for your correction, and I will review it again.
>
> **Q4**: ...would there be memory concerns if you are working with long-horizon time series forecasting / long context windows?
>
> **A4**: Your concerns are reasonable, and we have considered them. Therefore, we used a linear layer (shown on the right side of Figure 3) to reduce the length of the concatenated sequence, which facilitates subsequent computations. We will emphasize this point in the final version of the paper.
>
> [1]Yang S, Eisenach C, Madeka D. MQ-ReTCNN: Multi-Horizon Time Series Forecasting with Retrieval-Augmentation[J]. 2022.
> [2]Jing B, Zhang S, Zhu Y, et al. Retrieval-based time series forecasting[J]. arXiv preprint arXiv:2209.13525, 2022.
> [3] Zhang Y, Yan J. Crossformer: Transformer utilizing cross-dimension dependency for multivariate time series forecasting[C]//The eleventh international conference on learning representations. 2022.
> [4] Rasul K, Seward C, Schuster I, et al. Autoregressive denoising diffusion models for multivariate probabilistic time series forecasting[C]//International Conference on Machine Learning. PMLR, 2021: 8857-8868.

---

> > ### Comment · Reviewer_zh2V · 2024-08-11
> > **Thank you for your response**
> >
> > Thank you to the authors for addressing my questions.
> >
> > The answers to Q1, Q3, and Q4 have clarified my concerns, and I appreciate that the authors explicitly pointed out that lack of guidance mechanisms is one of the major barriers to the effectiveness of diffusion models in time series modeling, to further clarify the motivation behind this work.
> >
> > Maybe I did not make myself clear for Q2, what I intended to ask was why diffusion models would be preferred for certain time series modeling tasks over other non-diffusion model based methods, rather than the disadvantages of current diffusion models for time series. For example, is it because denoising would make the predictions more generalizable or more flexible, compared to other non-diffusion model based time series forecasting methods?
> >
> > Overall, this paper is interesting to me and I believe it will provide insights to future research in this direction. I keep my score.

---

> > > ### Author Response · Authors · 2024-08-11
> > > **Thank you for your reply**
> > >
> > > Thank you very much for your reply!
> > >
> > > I'm glad that our answer could address some of your concerns. Regarding Q2, I believe the main advantage of diffusion models stems from their framework's ability to learn precise distributions (the multi-stage framework reduces the difficulty of learning from prior distributions to data distributions). Learning the accurate conditional distribution for time series forecasting tasks can improve forecasting accuracy.
> > >
> > > Thank you for your recognition of our work! If you have any further questions, feel free to ask.

---

> > > > ### Comment · Reviewer_zh2V · 2024-08-12
> > > > **Thank you for your reply**
> > > >
> > > > Thanks for the reply and sharing the insights! It makes sense to me.

---

### Official Review · Reviewer_FgUz · 2024-07-11

**Soundness:** 3
**Presentation:** 3
**Contribution:** 3
**Rating:** 7
**Confidence:** 4

**Summary:**

This paper introduces the Retrieval-Augmented Time series Diffusion model (RATD) to address the limitations of existing time series diffusion models, such as insufficient datasets and lack of guidance. RATD consists of two components: an embedding-based retrieval process and a reference-guided diffusion model. The model retrieves the most relevant time series from a database to guide the denoising process, thereby enhancing the utilization of the dataset and providing meaningful guidance. Experiments on multiple datasets demonstrate the effectiveness of RATD, particularly in complex prediction tasks.

**Strengths:**

- **Novel Approach**: The idea of leveraging retrieved similar historical time series to guide forecasting for current timesteps is novel. References can potentially accelerate the convergence of the diffusion process and improve prediction accuracy.
- **Guidance Mechanism**: The reference-guided mechanism compensates for the lack of guidance in existing time series diffusion models, which is a significant contribution to the field.

**Weaknesses:**

- **Incomplete Experimental Comparisons**: The experimental setup and comparisons are insufficient and not aligned with existing studies, making it difficult to verify the performance and correct implementation of this work. For example, the datasets and forecasting horizons used differ from those in classical studies like CSDI and TimeGrad. This discrepancy makes it challenging to ensure fair comparisons and reproducibility.
  - In CSDI and TimeGrad, solar, electricity, traffic, taxi, wiki datasets and forecasting horizon=24 are used for comparisons. While this work adopts datasets of exchange, wind, electricity, and weather and only report the average results of horizons of 96, 192, and 336.
  - Given that the code is also not provided at the current stage, though I like the idea, I am not sure about the reproducibility. Hence I suggest the author to align with existing classical studies and provide a more comprehensive experimental results.
- **Insufficient Ablation Tests**: While Table 3 indicates the importance of choosing good retriever embeddings, there are no systematic ablation tests of the architecture developed (Figure 3). Detailed explanations of model architectures and extensive ablation tests are crucial to understanding the success of conditional diffusion and ensuring computational efficiency. The current paper lacks this information. It is easy for me to understand the conditional diffusion process. but it is extremely challenging for me to understand the insights and tricks in the architectures shown in Figure 3. I believe these are crucial to ensure the success of conditional diffusion and provide efficient computation. I cannot find any contents in the main paper or appendix informing me about these.
  - Detailed explanation of model architectures and extensive ablation tests are important for me to effectively distinguish it from CSDI. For example, you can still follow the architecture of CSDI while only introducing the depdence on retrived references x^R.
- **Unknown GPU Memory Usage**: There is no information on GPU memory usage and computation efficiency. Understanding how the introduction of reference series impacts computation and how these challenges are addressed by specific architectural designs is important for evaluating the practicality of the model.

**Questions:**

- More comprehensive and detailed comparisons with existing studies
  - Covering necessary datasets and both short and long horizons
- Systematic ablation tests to illustrate your specific architecture designs
  - Especially for Figure 3, more explanations and ablation tests are indispensable for readers to catch up

**Limitations:**

The idea of retrieval-augmented diffusion time-series model is interesting and reasonable.
My major concerns are about the incomplete experiments and the reproducibility.

I may consider raising the score if my major concerns are properly addressed.

---

> ### Author Rebuttal · Authors · 2024-08-06
>
> *Thank you very much to reviewer FgUZ for recognizing our proposed method. Following your suggestion, we have conducted additional experiments and provided further explanations of the model's structure. We hope to receive your approval.*
>
> **Q1**: The experimental setup and comparisons are insufficient and not aligned with existing studies, making it difficult to verify the performance and correct implementation of this work.
>
> **A1**: We extended the horizon length in our experiments, which is currently a reasonable practice as many popular forecasting methods [1][2] evaluate results by predicting horizon lengths of 96 and above. Of course, we agree with your point, so we supplemented the experiments according to the settings of CSDI, and the results are as follows.
>
> |Dataset | Traffic |  Electricity | Wind |
> | :-----| :---- | :---- |:---- |
> |Metric | MSE CRPS-sum |  MSE CRPS-sum | MSE CRPS-sum |
> | CSDI [3]| 0.027 0.020 |  0.112 0.017 | 0.462 0.042|
> | TimeGrad [4]| 0.041 0.044 |  0.121 0.021 |0.491 0.047 |
> | RATD(Ours)|0.023 0.018 |  0.096 0.014 | 0.311 0.031 |
>
> We compared CSDI, TimeGrad, and RATD under the experimental setting of CSDI (with a prediction window length of 24). It is noteworthy that in this experimental setup, the differences in results among different methods were narrowed. This may be due to the reduced difficulty of the task with a smaller prediction window. From the extra experiment, our method still exhibits obvious advantages.
>
>  **W2**: Insufficient Ablation Tests: While Table 3 indicates the importance of choosing good retriever embeddings, there are no systematic ablation tests of the architecture developed (Figure 3).
>
>  **A2**: Thank you very much for your suggestions! I truly appreciate that you took the time to read our paper thoroughly and provided very constructive feedback! To validate the ideas you mentioned, we conducted additional ablation experiments and performed preliminary tests on three datasets. The results are shown in the table below:
>
> |Dataset | Exchange|  Electricity | Wind |
> | :-----| :---- | :---- |:---- |
> | CSDI [3]| 0.077  |  0.379 | 1.066 |
> | CSDI+Linear| 0.075 |  0.316  | 0.932  |
> | CSDI+Cross Attention| 0.028  |  0.173 |0.829 |
> | CSDI+RMA|0.013 |  0.151  | 0.784 |
>
> Where CSDI represents the most basic network framework, CSDI+Linear denotes the approach where inputs and references are concatenated via a linear layer and fed into the network together (following [6]), CSDI+CrossAttention signifies the use of cross attention to fuse features from inputs and references (following [5]), and finally, CSDI+RMA, which incorporates an additional Reference Modulated Attention (RMA).
>
> Through our experiments, we found that compared to the basic Cross-attention-based method, RMA can integrate an edge information matrix (representing correlations between time and feature dimensions) more effectively. The extra fusion is highly beneficial in experiments, guiding the model to capture relationships between different variables. In contrast, linear-based methods concatenate inputs and references initially, which prevents the direct extraction of meaningful information from references, resulting in comparatively modest performance.
>
>  **W3**: There is no information on GPU memory usage and computation efficiency...
>
>  **A3**: As mentioned in our conclusion and limitation section, the retrieval process incurs additional computational costs. To minimize this cost, we pre-retrieve references for all samples during training and record the corresponding indices (as noted in line 215 of the paper). This means that during actual training and inference, we only need to consider the additional cost introduced by RMA. Since you are concerned about this aspect, we provide a rough overview of GPU memory usage and computational efficiency (all the experimental settings are the same as the paper, we complete the comparison on the wind dataset).
> || GPU memory usage (training)|  Time Cost (for test batch)|
> | :-----| :---- | :---- |
> | CSDI | ~11.1 (GB) |  ~240s |
> | RATD (ours)| ~12.3 (GB) | ~270s  |
>
> In general, compared to CSDI, our method has a slight disadvantage in terms of memory consumption and computational efficiency. However,  this disadvantage is not significant and does not pose serious challenges to applications.
>
>  **Q4**: More comprehensive and detailed comparisons with existing studies...
>
>  **A4**: We have provided supplementary experimental results above in hopes of addressing your concerns. Thank you.
>
>  **Q5**: Systematic ablation tests to illustrate your specific architecture designs...
>
>  **A5**: Similarly, we hope our ablation experiments have addressed your concerns. If you have any further questions, please feel free to reply. Thank you.
>
>
> [1]Liu Y, Hu T, Zhang H, et al. iTransformer: Inverted Transformers Are Effective for Time Series Forecasting[C]//The Twelfth International Conference on Learning Representations.
> [2]Nie Y, Nguyen N H, Sinthong P, et al. A Time Series is Worth 64 Words: Long-term Forecasting with Transformers[C]//The Eleventh International Conference on Learning Representations.
> [3]Yusuke Tashiro, Jiaming Song, Yang Song, and Stefano Ermon. Csdi: Conditional score-based diffusion models for probabilistic time series imputation. Advances in Neural Information Processing Systems, 34:24804–24816, 2021.
> [4]Rasul K, Seward C, Schuster I, et al. Autoregressive denoising diffusion models for multivariate probabilistic time series forecasting[C]//International Conference on Machine Learning. PMLR, 2021: 8857-8868.
> [5]Yang S, Eisenach C, Madeka D. MQ-ReTCNN: Multi-Horizon Time Series Forecasting with Retrieval-Augmentation[J]. 2022.
> [6]Jing B, Zhang S, Zhu Y, et al. Retrieval based time series forecasting[J]. arXiv preprint arXiv:2209.13525, 2022.

---

> > ### Comment · Reviewer_FgUz · 2024-08-08
> > **Thank you for your response**
> >
> > I appreciate that the authors have done some additional experiments to address my questions.
> >
> > This work looks interesting to me. But I do have some remaining concerns regarding insufficient experiments.
> >
> > First, why some datasets are still missing, such as solar and wiki in CSDI [1]. According to a recent benchmark study [2], solar is pretty special as it includes complex data distribution to be captured.
> >
> > [1] https://arxiv.org/pdf/2107.03502
> > [2] https://arxiv.org/pdf/2310.07446
> >
> > Second, regarding main experiments and ablation tests, a comprehensive experiments covering diverse datasets (as mentioned above), varied horizons, and evaluation metrics would convince me about the solidness of this work.
> > - About the combation of datasets and horizons, the original paper covers some and additional experiments in response gives some but still do not deliver a comprehensive set of experiments
> > - Ablation tests should be aligned with the evaluation scenarios of main experiments. And some analysis to explain the performance gains would be helpful.
> >
> > In summary, I like this work's idea but its current experimental results do not convince me sufficiently. So I keep my score.

---

> > > ### Author Response · Authors · 2024-08-08
> > > **Thank you for your reply**
> > >
> > > We are happy to see your reply!
> > >
> > > First, we could not complete experiments on all datasets before the first author rebuttal deadline due to the time required for the preprocessing of the dataset (including the retrieval process) and training. Therefore, we only presented partial results and hope for your understanding.  Currently, training on the remaining datasets is still ongoing, and we are about to obtain comprehensive experimental results.
> > >
> > > Second, we will follow your suggestion to conduct more comprehensive ablation experiments to demonstrate the effectiveness of the proposed new module.
> > >
> > > We expect to have all experimental results ready for presentation by this time tomorrow. Once again, thank you for your reply, and we will showcase the complete additional experiments tomorrow. Thank you！

---

> ### Author Response · Authors · 2024-08-09
> **More Experiment Results**
>
> Hello! Here we provide more comprehensive experimental results to address any concerns you may have about the completeness of our experiments. Our experimental findings are as follows, with additional experiments on all datasets mentioned in CSDI using identical experimental setups to CSDI(CRPS means CRPS-sum here).
>
> |Dataset | Traffic|  Electricity | Wind | Wiki | Taxi| Solar | Mimic-IV|
> | :-----| :---- | :---- |:---- | :---- |:---- |:---- |:---- |
> |Metric | MSE CRPS |  MSE CRPS | MSE CRPS | MSE CRPS | MSE CRPS | MSE CRPS | MSE CRPS |
> | CSDI |0.027 0.020|0.112 0.017|0.462 0.531|0.057 0.047|0.262 0.123|0.381 0.298|0.201 0.261|
> | TimeGrad | 0.041 0.044 |  0.121 0.021 |0.491 0.569 | 0.059 0.049| 0.253 0.114|0.359 0.287| 0.217 0.279|
> | RATD(Ours)|**0.023 0.018** |  **0.096 0.014** | **0.311 0.471** | **0.041 0.044**| **0.191 0.109**|**0.301 0.251**| **0.153 0.253**|
>
> From the results, it can be observed that our method performs better across all datasets. To quantify this more explicitly, we have calculated the average performance improvement （API） of our method compared to CSDI under both experimental settings.
>
> | Dataset | Traffic |  Electricity | Wind | Wiki | Taxi | Solar | Mimic-IV |
> | :-----| :---- | :---- |:---- | :---- |:-----|:---- |:-----|
> | API (%) in CSDI Settings |17.39|14.28|20.35|17.91|19.19|18.35|13.41|
> | API (%) in Our Settings |31.21|33.18|28.45|21.06|26.14|31.24|25.12|
>
>
> It appears that our method will indeed achieve better performance on more complex datasets. Notably, our method exhibits higher performance gains in more complex experimental setups, aligning with our motivation to tackle more intricate time series forecasting tasks.
>
> Additionally, we have supplemented results from disintegration experiments on six of these datasets.
> |Dataset | Exchange|  Electricity | Wind | Weather |Solar |Mimic-IV|
> | :-----| :---- | :---- |:---- | :---- |:---- |:---- |
> | CSDI| 0.077  |  0.379 | 1.066 |0.356|0.381|0.268|
> | CSDI+Linear| 0.075 |0.316|0.932| 0.349|0.369|0.265|
> | CSDI+Cross Attention| 0.028  |  0.173 |0.829 |0.291|0.340|0.183|
> | CSDI+RMA|**0.013** |  **0.151**  | **0.784** | **0.281**|**0.327**|**0.172**|
>
> As mentioned earlier, CSDI represents the basic network framework, CSDI+Linear denotes the approach where inputs and references are concatenated via a linear layer and fed into the network together, CSDI+CrossAttention signifies the use of cross attention to fuse features from inputs and references, and finally, CSDI+RMA, which incorporates an additional Reference Modulated Attention (RMA).
>
> For further analysis of RMA performance, we have also supplemented additional qualitative and quantitative experiments to compare the similarity between reference and prediction results. Due to our current inability to upload images or links, we present quantitative experimental results in the table below. Specifically, we use the Mean Squared Error (MSE) metric to measure the similarity between the reference and predicted results (where smaller values indicate greater similarity). Additionally, to assess whether the model effectively utilizes explicit information from the reference rather than noise, we employ the STL(
> Seasonal-Trend decomposition using LOESS) method to evaluate the similarity of different components. The results are as follows:
>
> |Dataset: Solar | Overall Similarity| Trend Component Similarity |Seasonal Component Similarity |
> | :-----| :---- | :---- |:---- |
> | CSDI+Linear| 0.986 |0.401|0.392|
> | CSDI+Cross Attention| 0.533 | 0.201 |0.264 |
> | CSDI+RMA|**0.471** |  **0.193**  | **0.241** |
>
> |Dataset: Mimic-IV | Overall Similarity| Trend Component Similarity |Seasonal Component Similarity |
> | :-----| :---- | :---- |:---- |
> | CSDI+Linear| 0.811 |0.381|0.202|
> | CSDI+Cross Attention| 0.401 | 0.191 |0.143 |
> | CSDI+RMA|**0.305**|  **0.142**  |**0.091** |
>
>
> In some complex time series datasets (Solar), trend components are more important for predictions, while for ECG sequences (Mimic-IV), periodicity is more crucial. Due to enhanced feature integration capabilities, our method adaptively captures the most important information for both cases, leading to better prediction results. In summary, our approach efficiently captures information from the reference and provides beneficial guidance on periodicity and trends for the generated time series.
>
> Finally, we sincerely hope that the additional experiments we propose can address your concerns regarding the completeness of the experiments! If you have any further questions, please let us know!

---

> > ### Comment · Reviewer_FgUz · 2024-08-09
> > **Thank you for adding additional experiments**
> >
> > I appreciate the efforts the authors have taken to address my concerns.The results look good to me. My last question would be are you planning to open source your code? As I feel that such a retrieval-based diffusion may not be easy to reproduce. If possible, could you share some anonymous implementation during the review period?

---

> > > ### Author Response · Authors · 2024-08-09
> > > **Thank you for your reply**
> > >
> > > Thank you very much for your reply! We are glad to see that some of your concerns have been addressed.
> > >
> > > I am happy to share our code with you here through an anonymous GitHub link, but the conference policy does not permit posting links or similar actions. If our paper is accepted, we will immediately make the Github link available in the paper. Even if this process requires a formal application, we will upload and disclose the code as soon as possible. We hope for your understanding.
> > >
> > > Thanks again for your reply! Please feel free to ask any further questions.

---

> > > > ### Comment · Reviewer_FgUz · 2024-08-09
> > > > **LGTM**
> > > >
> > > > I trust your promise to share code and ensure the reproducibility. Most of my concerns have been addressed properly. So I have raised the score.

---

> > > > > ### Author Response · Authors · 2024-08-09
> > > > > **Thank you for your support**
> > > > >
> > > > > Thank you very much for raising the score!
> > > > >
> > > > > We greatly appreciate your suggestions and support for our work. We will carefully consider incorporating your recommendations to enrich the experimental content in the final version of the paper.
> > > > >
> > > > > If you have any further questions, please feel free to reply. Thank you!

---

### Official Review · Reviewer_uLZU · 2024-07-11

**Soundness:** 3
**Presentation:** 2
**Contribution:** 2
**Rating:** 4
**Confidence:** 3

**Summary:**

This article proposes a retrieval-augmented diffusion model for time series prediction, featuring a simple and straightforward approach. It aims to address two key issues:

1. The lack of semantics and labels in time series data, leading to insufficient guidance during the diffusion model's generation process.

2. The problem of size insufficiency and imbalance in time series data.

Specifically, the method involves embedding-based retrieval to gather similar time series data from a database to serve as references during the denoising process. Their proposed RMA module performs feature fusion between the retrieved time series and the current diffusion step's results, thereby enhancing the denoising and generation process of the diffusion model.

In the experimental section, they explore the impact of different retrieval methods on the performance of the diffusion model. They also investigate the effects of varying the size of the retrieval database and the number of retrieved sequences (k) on the prediction performance.

**Strengths:**

An additional database was utilized to enhance the diffusion generation process. This method is straightforward and easy to follow. The newly proposed RMA module aids in integrating features from different datasets. The paper is clearly written and achieves state-of-the-art (SOTA) results on the majority of the datasets they selected.

**Weaknesses:**

The paper lacks innovation as the retrieval-enhanced diffusion method has already been proposed in the field of image generation, and similar retrieval-enhanced approaches exist in the time-series prediction domain, although their backbone is not a diffusion model. Additionally, the performance improvement largely stems from the representational capabilities of the pre-trained embedding model used.

Typo in title: "Augment" -> "augument".

**Questions:**

1. What would be the result if the retrieved sequences were directly used for a simple weighted average for the final generation output, or if the performance of selecting the sequence closest to the ground truth (gt) from the retrieved ones was evaluated?
2. I am uncertain whether the non-diffusion time-series prediction methods used in the author's comparative experiments include retrieval-enhanced modules. If not, I recommend adding experiments with this component to demonstrate the efficacy of the proposed method.
3. In Formula 1, the symbol $\mathcal{D}$ is used ambiguously, representing both distance and feature dimensions.

**Limitations:**

The approach may heavily rely on the capabilities of the embedding model and the construction of the database.

---

> ### Author Rebuttal · Authors · 2024-08-06
>
> *Thank you very much to reviewer uLZU for acknowledging some advantages of our proposed method. We understand you may have concerns regarding the overall novelty of the entire paper. We will address this point.*
>
> **Q1**: The paper lacks innovation as the retrieval-enhanced diffusion method has already been proposed in the field of image generation...
>
> **A1**： The innovation of this method is reflected in the following aspects：
> * We **for the first time utilize RAG on time series diffusion models**. In text2image diffusion models, cross-modal guidance mechanisms are well-established, and retrieval-enhanced methods emphasize using semi-parametric approaches to enhance generation efficiency. However, time series diffusion models lack a universally recognized effective guidance mechanism.  **We are the first to provide a promising RAG-based guidance mechanism for the diffusion generation of time series, which represents a pioneering advancement in time series diffusion model research**. Additionally, while existing work utilizes retrieval mechanisms for time series prediction, previous efforts have not adequately analyzed the use of references nor extensively evaluated and compared model performance in experiments. Our novel framework addresses these gaps in previous research and significantly advances the study of retrieval-enhanced time series prediction methods.
> * Our method **introduces a novel attentional mechanism for time series diffusion, called RMA (Reference Modulated Attention, L167-180)**, designed to enhance the guidance provided by references in the generation process. While the overall structure is not complex, our proposed RMA is effective and similarly groundbreaking. Additionally, we designed extra ablation experiments to demonstrate the effectiveness of RMA.
>  |Dataset | Exchange|  Electricity | Wind |
> | :-----| :---- | :---- |:---- |
> | CSDI | 0.077  |  0.379 | 1.066 |
> | CSDI+Linear| 0.075 |  0.316  | 0.932  |
> | CSDI+Cross Attention| 0.028  |  0.173 |0.829 |
> | CSDI+RMA|0.013 |  0.151  | 0.784 |
>
> Where CSDI represents the most basic network framework, CSDI+Linear denotes the approach where inputs and references are concatenated via a linear layer and fed into the network together, CSDI+CrossAttention signifies the use of cross attention to fuse features from inputs and references, and finally, CSDI+RMA, which incorporates an additional  RMA. Clearly, RMA is highly effective in leveraging references for guidance.
>
> Regarding the issue of dependency on pre-trained encoders, we discussed this in Section 5.3 (line 240). In the embedding-based retrieval mechanism, different encoders (feature extractors) do indeed introduce performance differences. However, these differences are not decisive. This variability may stem from the effectiveness of the methods we selected and the quality of the retrieved references. This also indicates that our method exhibits high robustness and does not rely excessively on the choice of pre-trained encoders.
>
> **Q2**: There is a typo in the last sentence of the second paragraph from Section 5.2. “.t”
>
> **A2**: I apologize for the typo. It was an oversight on our part, and I will review the paper again. Thank you for bringing it to my attention.
>
> **Q3**: What would be the result if the retrieved sequences were directly used for a simple weighted average for the final generation output...
>
> **A3**: Your suggestion is very interesting! Indeed, we paid close attention to this issue during our experimental phase. If the references are too similar to the ground truth (GT), the model's predictions can overly rely on the references, thereby failing to learn the true conditional distribution. We assessed this visually during experiments, such as in our analysis in Figure 4 from the paper. Following your advice, we conducted experiments on two datasets.
>
> |Dataset |Exchange| Wind |
> | :---- | :---- | :---- |
> |RATD（Ours）  | 0.013 | 0.784  |
> |Reference (Closest)| 0.153 | 2.487|
> |Reference (Average)| 0.197| 2.597 |
>
> Specifically, "Reference (Closest)" refers to the closest reference. The results demonstrate that on datasets with strong periodic patterns, the references and predicted results are indeed very similar, whereas, on more complex datasets(wind), the references cannot replace the prediction at all.
>
> **Q4**: I am uncertain whether the non-diffusion time-series prediction methods used in the author's comparative experiments include retrieval-enhanced modules...
>
> **A4**: We did not compare our method with the previous RAG-based time series methods in the paper because these papers did not provide comprehensive evaluations on commonly used time series datasets or publicly available code. The implementation details in these papers were also very limited, making replication of their methods challenging. Displaying replicated results directly in the main text could potentially lead to unnecessary misinterpretation. Nevertheless, as you mentioned, for the sake of comprehensive experimental comparison, we present here the MSE results of our replicated method under our experimental settings, demonstrating that our approach shows noticeable advantages.
>
> | |Exchange| Wind | Traffic | Weather|
> | :-----| :---- | :---- |:---- |:---- |
> |RATD（Ours）  | 0.013(0.001)  | 0.784(0.005)  | 0.151(0.002) |0.281(0.002) |
> |MQRetNN| 0.063(0.004) | 1.116(0.008) |0.346(0.003) |0.668(0.004) |
> |ReTime| 0.059(0.003) | 1.043(0.007) |0.330(0.005)|0.489(0.005)|
>
> Clearly, RATD exhibits significant performance advantages, indicating that our method can better utilize references for forecasting.
>
> **Q5**: In Formula 1, the symbol D is used ambiguously, representing both distance and feature dimensions.
>
> **A5**: I apologize for the mistake in using symbols. I will correct the paper and remove any potentially misleading parts.

---

> > ### Author Response · Authors · 2024-08-12
> > **Awaiting your reply**
> >
> > Hello, Reviewer uLZU,
> > Regarding your concerns, we have provided some responses that you might find useful, including:
> > * We have highlighted and summarized the innovations of our method from two perspectives: innovation in model design and experimental results.
> > * Additional experimental results, including treating references as predictions and comparing our proposed method with previous reference-based methods.
> > * We have explained that the performance of our method is not highly dependent on the pre-trained encoding models.
> >
> > We sincerely hope our responses address some of your concerns. If you have any further questions, please feel free to ask. Thank you.

---

> > > ### Comment · Reviewer_uLZU · 2024-08-12
> > > **Thank you for your response.**
> > >
> > > The author has addressed the review comments thoroughly, and most of my questions have been answered satisfactorily.
> > >
> > > However, I still have concerns regarding the novelty of the proposed method, a point also raised by Reviewer zh2V. While the author highlights their contributions, such as being the first to propose a retrieval-enhanced diffusion model for time series and incorporating an attention mechanism, I remain uncertain whether the solution effectively addresses the core challenges associated with diffusion models in time series forecasting. Nevertheless, the integration of techniques like retrieval and diffusion modeling is an interesting and promising approach for this area.
> > >
> > > Regarding Q4, the base structures of the methods being compared are not built on diffusion models. Could you clarify the specific advantages introduced by your base model or the retrieval-augmented solution?

---

> > > > ### Author Response · Authors · 2024-08-13
> > > > **Awaiting your reply**
> > > >
> > > > Dear Reviewer uLZU:
> > > >
> > > > Should there be any further points that require clarification or improvement, please know that we are fully committed to addressing them promptly.
> > > >
> > > > Thank you once again for your invaluable contribution to our research.
> > > >
> > > > Warm Regards,
> > > >
> > > > The Authors

---

> ### Author Response · Authors · 2024-08-12
> **Thank you for you reply**
>
> We are glad to receive your reply! We are also pleased that most of your concerns have been addressed.
>
> Regarding your concerns about the paper's novelty, I would like to provide some additional clarifications. In addition to the lack of an effective guiding mechanism of exisiting time series diffusion models, another core issue is the limited size or quality of existing datasets. Specifically, real-world time series datasets are either too small in scale or highly imbalanced, which may not meet the high-quality data requirements of training diffusion models[1]. Our approach offers a general solution by leveraging useful information from the dataset during the generation process progressively, rather than expending significant resources to augment the existing dataset. In other words, our method makes the most of the limited dataset, thereby addressing the aforementioned core issue to some extent.
>
>  Additionally, a critical challenge in time series forecasting is modeling both the periodic and trend components of the time series simultaneously [2]. Our approach provides substantial assistance in direct and explicit modeling of these temporal components by utilizing guidance from references.
>
>  Overall, our method focuses on and addresses these core issues, making it both innovative and practical, which have been recognized by other reviewers.
>
> Regarding the supplementary experiments for A4, the performance advantages of our method stem from three aspects:
> * **Advantages of Diffusion Model Framework**:  Unlike methods that directly use a transformer, the multi-stage framework of diffusion models reduces the difficulty of learning from prior distributions to data distributions, which helps the model learn more accurate conditional distributions.
> * **Iterative Guidance Mechanism**: Our effective guidance mechanism allows the utilization of conditions (i.e., references) during the diffusion process iteratively (T =100 in our experiment), whereas previous methods only performed feature fusion once. In other words, diffusion models can leverage references as guidance more effectively.
> * **Better Feature Fusion Module**: The proposed RMA (Retrieval-Modulated Attention) module performs better in feature fusion compared to the previous work. We designed an extra experiment to prove it (results are shown in the previous response A1). Our experiments found that compared to the basic cross-attention-based method (used by ReTime) and linear fusion methods (used by MQ-ReTCNN), RMA can integrate an extra information matrix (representing correlations between time and feature dimensions), thus realizing the fusion of feature more effectively.
>
> In summary, the first two advantages stem from the diffusion model framework itself, while the last advantage comes from the new module we proposed. By integrating the three components, our proposed method achieved better experimental results.
>
> Once again, thank you for your reply. We are so honored that you find our work interesting and promising. If you have any further questions, please feel free to ask. Thank you!
>
> [1] Schramowski P, Brack M, Deiseroth B, et al. Safe latent diffusion: Mitigating inappropriate degeneration in diffusion models[C]//Proceedings of the IEEE/CVF Conference on Computer Vision and Pattern Recognition. 2023: 22522-22531.
>
> [2]Shen L, Chen W, Kwok J. Multi-Resolution Diffusion Models for Time Series Forecasting[C]//The Twelfth International Conference on Learning Representations. 2024.

---

### Official Review · Reviewer_AGBG · 2024-07-12

**Soundness:** 3
**Presentation:** 2
**Contribution:** 2
**Rating:** 5
**Confidence:** 3

**Summary:**

Existing time series diffusion models are unstable due to insufficient datasets and lack of guidance. The RATD model combines an embedding-based retrieval process with a reference-guided diffusion model to improve stability and accuracy. RATD retrieves relevant time series from a database to guide the denoising process, maximizing dataset utilization and compensating for guidance deficiencies. Experiments on multiple datasets demonstrate RATD's effectiveness, particularly in complex prediction tasks.

**Strengths:**

- The paper introduces a creative combination of embedding-based retrieval and reference-guided diffusion. This approach is innovative in addressing the limitations of existing time series diffusion models.
- The authors conducted extensive experiments on multiple datasets, demonstrating the effectiveness of RATD in complex prediction tasks. The results show that RATD outperforms existing methods in terms of stability and accuracy.
The paper provides a thorough explanation of the model architecture, retrieval mechanism, and training procedure, ensuring reproducibility.
- The paper is well-organized, with clear sections on introduction, related work, methodology, experiments, and conclusions. The use of figures and tables enhances understanding.
- The authors explain complex concepts in a concise manner, making the paper accessible to readers with varying levels of expertise.

**Weaknesses:**

- The proposed RATD lacks justification for architecture.
- Please refer to the below questions.

**Questions:**

- What is the main difference between RAG and proposed work?
- How to define stability in time-series data? Someone may not agree with stability in Figure 1-(c). Some may disagree that the red plots are unstable.
- Why necessarily the proposed RATD is needed in time-series forecasting? In other words, how to link the relationship between RATD and TSAD?
- There is a typo in the last sentence of the second paragraph from Section 5.2. “.t”
- Why Table 2 is separated from Table 1 independently?
- What can you define as the best retrieval for forecasting?

**Limitations:**

They described their limitations themselves.

---

> ### Author Rebuttal · Authors · 2024-08-06
>
> *We thank Reviewer AGBG for the thorough and valuable feedback. We are glad that the reviewer found that the proposed model is effective and our paper is easy to read. The main concern of the reviewer is that the architecture of RATD lacks justification. Please see below for our responses to your concerns.*
>
> **Q1**: The proposed RATD lacks justification for architecture.
>
> **A1**: We here justify the effectivness of our architecture from following aspects:
>
> * We **for the first time** introduce the concept of RAG to time series diffusion models. In other words, and **introduces a new guiding mechanism** for time series diffusion models.  Describing time series data directly is challenging, and currently, there is no universally recognized effective guidance mechanism that guarantees the effectiveness of diffusion models in time series conditional generation tasks.
> * The reference-augmented mechanism we propose is highly effective.  Our model framework is based on CSDI (mentioned in Line 167), and building upon CSDI, our approach achieves superior performance. We have listed the performance improvement ratios (MSE) in the table below.
>
> | | Exchange |  Electricity | Wind | Weather |
> | :-----| :---- | :---- |:---- | :---- |
> | Performance improvement (%) | 79.22 |  60.18 | 26.45 | 21.06 |
>
> * The **RMA (Reference Modulated Attention, L167-180)** we propose is also robust and effective. In terms of robustness, the performance does not rely excessively on pre-trained encoders（from line 250 on paper). Additionally, we designed extra ablation experiments to demonstrate the effectiveness of RMA (Metric: MSE).
> |Dataset | Exchange|  Electricity | Wind |
> | :-----| :---- | :---- |:---- |
> | CSDI | 0.077  |  0.379 | 1.066 |
> | CSDI+Linear| 0.075 |  0.316  | 0.932  |
> | CSDI+Cross Attention| 0.028  |  0.173 |0.829 |
> | CSDI+RMA|0.013 |  0.151  | 0.784 |
>
> Where CSDI represents the most basic network framework, CSDI+Linear denotes the approach where inputs and references are concatenated via a linear layer and fed into the network together, CSDI+CrossAttention signifies the use of cross attention to fuse features from inputs and references, and finally, CSDI+RMA, which incorporates an additional RMA. Clearly, RMA is highly effective in leveraging references for references.
>
> **Q2**: What is the main difference between RAG and the proposed work?
>
> **A2**: In our related work (section 2.2, line 76), we mentioned RAG and positioned our proposed RATD as a new application of RAG in the study of time series diffusion models. Compared to previous time series RAG methods, our approach still holds significant advantages. This advantage stems from the iterative structure of the diffusion model and our proposed Reference Modulated Attention, where references can repeatedly influence the generation process, allowing references to provide a informative guidance for the conditional generation process.
>
> **Q3**: How to define stability in time-series data? Someone may not agree with stability in Figure 1-(c)...
>
> **A3**: To validate our conclusions about 'stability,' we designed additional experiments. In these experiments, we calculated the variance of 15 repeated prediction results, which are presented in the table below. By evaluating the variance, we found that our method demonstrates a clear advantage in stability through the use of additional references for guidance.
>
> | |Exchange| Wind | Traffic | Weather|
> | :-----| :---- | :---- |:---- |:---- |
> |RATD（Ours）  | 0.013(0.001)  | 0.784(0.005)  | 0.151(0.002) |0.281(0.002) |
> |iTransformer| 0.016(0.001) | 0.932(0.007) |0.192(0.003) |0.358(0.003) |
> |PatchTST| 0.047(0.009) | 1.001(0.009) |0.225(0.003)|0.782(0.008)|
> |CSDI| 0.077(0.003) | 1.066(0.008) |0.379(0.003) |0.356(0.002)|
>
> **Q4**: Why necessarily the proposed RATD is needed in time-series forecasting? In other words, how to link the relationship between RATD and TSAD?
>
> **A4** : As we mentioned in **A1**, our method provides a new guiding mechanism for time series diffusion models, and it proves to be highly effective in experiments. Furthermore, our method is plug-in, allowing its application to other transformer-based time series diffusion models. The design principles and motivations behind RATD may also offer new insights for future work, particularly in designing novel guiding mechanisms. All of these aspects underscore the ‘necessity’ of our approach.
> If TSAD refers to Time Series Anomaly Detection, directly applying the retrieval process may not yield optimal results, as the anchors used for retrieval could be anomalous. However, with appropriate design, it might be possible to cleverly mitigate this issue, thereby making retrieval-based methods potentially viable for TSAD.
>
> **Q5**: There is a typo in the last sentence of the second paragraph from Section 5.2. “.t”
>
> **A5**: I apologize for the typo. I will correct the paper again. Thank you for bringing it to my attention.
>
> **Q6**: Why Table 2 is separated from Table 1 independently?
>
> **A6**: The main reason we separated Table 1 and Table 2 is due to our use of different strategies for constructing the retrieval databases (as mentioned in line 211). Additionally, most of the popular baselines have not yet been evaluated on the MIMIC-IV dataset, given its later release.
>
> **Q7**: What can you define as the best retrieval for forecasting?
>
> **A7**: Generally, a good pre-trained time series encoder can be a good retriever because it can effectively embed the trend of time series. In this paper, we leverage state-of-the-art pre-trained encoder models, and use them for retrieving the most similar anchor time series as reference. Such reference can naturally be the best retrieval for the conditional forecasting, which have been further proved by our empirical results in paper.

---

> > ### Author Response · Authors · 2024-08-12
> > **Awaiting your reply**
> >
> > Hi, Reviewer AGBG,
> > We have provided some responses that might address your concerns, including:
> > * We provided proof of the framework's justification, including additional experimental results and further analysis.
> > * We clarified the differences between our method and previous works while extra experiments demonstrate that our method has greater stability.
> > * We also offered some insights that you might be interested in about our proposed method and the related methods.
> >
> > We sincerely hope our responses address some of your concerns. If you have any further questions, please feel free to ask. Thank you.

---

> > > ### Comment · Reviewer_AGBG · 2024-08-12
> > >
> > > Thanks for your response, which addresses my concerns. I will raise my score.

---

> > > > ### Author Response · Authors · 2024-08-12
> > > > **Thank you for your reply**
> > > >
> > > > We are so glad to see your reply!
> > > >
> > > > Also, we are very grateful for your recognition of our work. If you have any further questions, please feel free to ask.
> > > >
> > > > Thank you!

---

### Official Review · Reviewer_HjSX · 2024-07-15

**Soundness:** 3
**Presentation:** 2
**Contribution:** 3
**Rating:** 5
**Confidence:** 3

**Summary:**

This paper proposes a retrieval-augmented time series diffusion model that uses an embedding-based retrieval process and a reference-guided diffusion model. The proposed RATD retrieves relevant time series from an external database as references, which is later utilized to guide the denoising process of the diffusion model for future forecasting. Extensive experiments are conducted on five real-world datasets that demonstrate the effectiveness of the proposed approach.

**Strengths:**

- The paper is clearly written and easy to understand. The authors provide clear and detailed explanations and illustrations for the proposed architecture.
- The idea of applying retrieval augmentation on the diffusion model for time series forecasting is novel and insightful.

**Weaknesses:**

- The proposed architecture requires retrieval database construction and pretraining the time series encoder $E_\phi$ before training the forecasting model. The additional complexity and computation cost may limit the efficiency of the proposed method.
- Experimental results do not contain standard deviation, which potentially limits the confidence and significance of the performance superiority.
- Some baselines are missing. For instance, aurhors mentioned MQ-ReTCNN and ReTime as retrieval-augemented time series models. However, they are not invovled as baselines in experiments.

**Questions:**

- How is the time series encoder $E_\phi$ pretrained with representation learning tasks (Line148)? Since the embedding quality highly affects the retrieval precision and quality, some analysis and evaluations should be conducted on the time series embedding quality.
- What is the architecture of the pre-trained encoder $E_\phi$? How do you encode the multivariate time series into a single embedding? Do you use any pooling operation? More analysis and explanations on the encoder are beneficial for paper clarity.

**Limitations:**

Yes. The authors discussed the limitations of the proposed approach in the paper.

---

> ### Author Rebuttal · Authors · 2024-08-06
>
> *We thank Reviewer HjSX for the thorough and valuable feedback. We are glad that the reviewer found that the proposed idea is novel. The reviewer's main concern is the use of pre-trained models. Please see below for our responses to your concerns.*
>
> **Q1**: The additional complexity and computation cost may limit the efficiency of the proposed method.
>
> **A1**: The demand for pre-trained models does indeed incur additional computational costs. However, the increased training costs do not impose severe constraints on the practical application of the model. The additional training costs stem from two components: pretraining the encoder and the retrieval process. Specifically, pretraining the encoder is straightforward, as we employ the structurally simple yet effective TCN. Typically, TCN can be trained on a dataset within a few hours on a standard single GPU (e.g., Nvidia RTX 3090), thereby avoiding excessively high time costs. Similarly, the entire retrieval process also incurs a few extra hours of computational time. Compared to the time costs required for training diffusion models (ranging from a day to several days), the extra cost is not a serious issue. Furthermore, it is worth noting that our model does not incur excessively high sampling costs (as depicted in Figure 6 of the original text), which is crucial because sampling costs of diffusion models are a more pertinent concern compared to training costs.
>
> **Q2**: Experimental results do not contain standard deviation, which potentially limits the confidence and significance of the performance superiority.
>
> **A2**: Thank you very much for your suggestion! Calculating the standard deviation will indeed further assess the stability of the model's generated results. Following your advice, we conducted additional experiments including some popular baselines, and the MSE results are shown in the table below (with the same experimental settings as in this paper, we repeat the test 15 times to calculate the standard deviation ).
>
> | |Exchange| Wind | Traffic | Weather|
> | :-----| :---- | :---- |:---- |:---- |
> |RATD（Ours）  | 0.013(0.001)  | 0.784(0.005)  | 0.151(0.002) |0.281(0.002) |
> |iTransformer| 0.016(0.001) | 0.932(0.007) |0.192(0.003) |0.358(0.003) |
> |PatchTST| 0.047(0.009) | 1.001(0.009) |0.225(0.003)|0.782(0.008)|
> |CSDI| 0.077(0.003) | 1.066(0.008) |0.379(0.003) |0.356(0.002)|
>
> Our method also demonstrates superior certainty in the results based on additional experiments.
>
> **Q3**: Some baselines are missing. For instance, the authors mentioned MQ-ReTCNN and ReTime as retrieval-augmented time series models.
>
> **A3**: We did not compare our method with the previous RAG-based time series methods in the paper because these papers did not provide comprehensive evaluations on commonly used time series datasets or publicly available codes. The implementation details in these papers were also very limited, making replication of their methods challenging. Nevertheless, as you mentioned, for the sake of comprehensive experimental comparison, we present here the MSE results of our replicated method under our experimental settings, demonstrating that our approach shows noticeable advantages.
> | |Exchange| Wind | Traffic | Weather|
> | :-----| :---- | :---- |:---- |:---- |
> |RATD（Ours）  | 0.013(0.001)  | 0.784(0.005)  | 0.151(0.002) |0.281(0.002) |
> |MQRetNN| 0.063(0.004) | 1.116(0.008) |0.346(0.003) |0.668(0.004) |
> |ReTime| 0.059(0.003) | 1.043(0.007) |0.330(0.005)|0.489(0.005)|
>
> RATD exhibits significant performance advantages, indicating that our method can better utilize references for prediction.
>
> **Q4**: How is the time series encoder Eϕ pre-trained with representation learning tasks (Line148)? Since the embedding quality highly affects the retrieval precision and quality, some analysis and evaluations should be conducted on the time series embedding quality.
>
> **A4**: We trained TCN for time series prediction tasks for pre-training. In Section 5.3（Influence of Retrieval Mechanism， line 240）of the paper, we extensively discussed the question of which encoder structure to adopt. I believe this discussion holds the same significance as the "discussion on embedding quality" you mentioned because the quality of embeddings is difficult to assess directly and can only be judged by comparing their contributions to experiment improvements. It is worth noting that through comparative experiments, we found that the differences in results brought by different encoders are not significant. This also demonstrates that our approach has strong robustness.
>
> **Q5**: What is the architecture of the pre-trained encoder Eϕ? How do you encode the multivariate time series into a single embedding? Do you use any pooling operation? More analysis and explanations on the encoder are beneficial for paper clarity.
>
> **A5**: Your concerns may be solved after reading Section 5.3（Influence of Retrieval Mechanism， line 240）. As we addressed above: we conducted a comprehensive discussion on the structure of the encoder. The TCN architecture supports the encoding of multivariate time series (resulting embeddings are not one-dimensional) without the need for additional pooling layers.

---

> > ### Comment · Reviewer_HjSX · 2024-08-13
> > **Thanks for your response.**
> >
> > Thanks for the response. The author rebuttal has addressed most of my concerns. I tend to maintain my score.

---

> > ### Author Response · Authors · 2024-08-14
> > **Thank you for your reply**
> >
> > Thank you for your reply.
> >
> > If you have further questions, please feel free to ask.

---

### Author Rebuttal · Authors · 2024-08-06

We sincerely thank all the reviewers for the thorough reviews and valuable feedback. We are glad to hear that the idea is novel or innovative (Reviewer HJsX, AGBG, and FgUz ), this paper is well-written and easy to follow (Reviewer HJsX, AGBG, uLZU and hz2v), and provide clear explanations for the proposed architecture (Reviewer HJsX, AGBG and hz2v).

Here, we want to highlight the novelty and contributions of our proposed method as follows:

* **First retrieval-augmented time series diffusion model**: We for the first time introduce a retrieval-augmented framework for time series diffusion model. The framework is concise, effective, and robust. RATD will inspire future research in time series diffusion models.
* **Specialized attention mechanism for reference-guided time series diffusion**: We introduce a new attentional mechanism (Reference Modulated Attention, RMA) to effectively utilize references for guiding the generation process. Compared to the cross-attention module, RMA more effectively integrates multiple features, making it a better for diffusion-based time series forecasting.
* **Comprehensive evaluation**: We evaluate our approach on five real-world datasets, consistently achieving state-of-the-art performance.

We summarized our responses to reviewers as follows:

* We analyzed the differences between our method and the existing time-series RAG method, and quantified these differences through experiments. (Reviewer HJsX, uLZU and hz2v)

* We supplemented the paper with ablation experiments for the newly proposed module (Reference Modulated Attention, RMA) to validate its effectiveness. (Reviewer AGBG and Fguz)

* By forecasting repeatedly, we obtained variance in the prediction processes for evaluation, providing the basis for model stability analysis. (Reviewer HJsX and uLZU)

* We provided more explanations for the innovation of the proposed method and the issue of the performance being overly reliant on pre-trained models. (Reviewer AGBG, uLZU and hz2v)

We reply to each reviewer's questions in detail below their reviews. Please kindly check out them. Thank you and please feel free to ask any further questions.

---

### Decision · Program_Chairs · 2024-09-25

**Decision:**

Accept (poster)

**Comment:**

This well written paper has been assessed by five knowledgeable reviewers. Four of them leaned towards accepting it (one full accept, one weak accept and two borderline accept ratings), while one leaned the other way (borderline rejection). The reviewers predominantly highlighted novelly of the presented approach and clarity of its presentation.  One of the reviewers expressed a dissenting opinion on the novelty of the proposed approach that is a clever combination of otherwise well established techniques. The authors provided a comprehensive rebuttal in response to the initial critique and engaged the reviewers in discussions, which all helped resolve most of the concerns. Given the prevailing sentiment, with which I agree, I conclude that this work satisfies the requirements for being presentable at NeurIPS.